# Unraveling the influence of non-fullerene acceptor molecular packing on photovoltaic performance of organic solar cells

Linglong Ye[1,2,11], Kangkang Weng[1,11], Jinqiu Xu[3,11], Xiaoyan Du[4,5,11], Sreelakshmi Chandrabose[6], Kai Chen[6], Jiadong Zhou[7], Guangchao Han[8], Songting Tan[2], Zengqi Xie[7], Yuanping Yi [8], Ning Li [4,5,9✉], Feng Liu [3✉], Justin M. Hodgkiss[6], Christoph J. Brabec [4,5] & Yanming Sun [1,10✉]

In non-fullerene organic solar cells, the long-range structure ordering induced by end-group π–π stacking of fused-ring non-fullerene acceptors is considered as the critical factor in realizing efficient charge transport and high power conversion efficiency. Here, we demonstrate that side-chain engineering of non-fullerene acceptors could drive the fused-ring backbone assembly from a π–π stacking mode to an intermixed packing mode, and to a non-stacking mode to refine its solid-state properties. Different from the above-mentioned understanding, we find that close atom contacts in a non-stacking mode can form efficient charge transport pathway through close side atom interactions. The intermixed solid-state packing motif in active layers could enable organic solar cells with superior efficiency and reduced non-radiative recombination loss compared with devices based on molecules with the classic end-group π–π stacking mode. Our observations open a new avenue in material design that endows better photovoltaic performance.

[1] School of Chemistry, Beihang University, 100191 Beijing, China. [2] Key Laboratory of Environmentally Friendly Chemistry and Applications of Ministry of Education, College of Chemistry, Xiangtan University, 411105 Xiangtan, China. [3] School of Chemistry and Chemical Engineering, Frontiers Science Center for Transformative Molecules, Shanghai Jiao Tong University, 200240 Shanghai, China. [4] Institute of Materials for Electronics and Energy Technology (i-MEET), Friedrich-Alexander-Universität Erlangen-Nürnberg, 91058 Erlangen, Germany. [5] Helmholtz-Institute Erlangen-Nürnberg for Renewable Energy (HI ERN), Immerwahrstr. 2, 91058 Erlangen, Germany. [6] MacDiarmid Institute for Advanced Materials and Nanotechnology, and School of Chemical and Physical Sciences, Victoria University of Wellington, Wellington 6010, New Zealand. [7] State Key Laboratory of Luminescent Materials and Devices, Institute of Polymer Optoelectronic Materials and Devices, Guangdong Provincial Key Laboratory of Luminescence from Molecular Aggregates, South China University of Technology, 510640 Guangzhou, China. [8] Beijing National Laboratory for Molecular Science, Key Laboratory of Organic Solids, Institute of Chemistry, Chinese Academy of Sciences, 100190 Beijing, China. [9] National Engineering Research Center for Advanced Polymer Processing Technology, Zhengzhou University, 450002 Zhengzhou, China. [10] Beijing Advanced Innovation Center for Biomedical Engineering, 100191 Beijing, China. [11]These authors contributed equally: Linglong Ye, Kangkang Weng, Jinqiu Xu, Xiaoyan Du. ✉email: ning.li@fau.de; fengliu82@sjtu.edu.cn; sunym@buaa.edu.cn

Non-fullerene small molecule acceptors (NFAs) that better harness long wavelength radiation and reduce unfavorable interfacial electron coupling lead to fundamental breakthrough in organic solar cells (OSCs) with a remarkable power conversion efficiency (PCE) of over 18%[1–8]. The well-defined molecular structure and readily controllable variables in chemistry constituted NFAs excellent semiconducting materials by design. The rigid geometry of NFA backbone and flexible side chain result in multiple handles that can be utilized to manipulate NFA solid-state electronic structure to better suit in photovoltaic function, as in results seen from the development of initiative ITIC family to the most recent Y6 analogues[9–12]. The properties of NFAs, including crystallization, molecular ordering and interaction, miscibility, etc. are important parameters that influence the nanostructure and optoelectronic properties of the resulting bulk-heterojunction (BHJ) blends[13–20]. π–π stacking in NFAs is thought as an important factor that controls charge transport, and thus the end-group stacking induced long-range order of NFAs is considered as a critical factor governing the merits of organic photovoltaics[21–24].

Multiple molecular packing motifs are regarded as essential in guiding different transport channels in small molecule organic semiconductors, as seen from previous research of polyaromatic hydrocarbons or thioacences that herring bond structure, one-dimensional slip stack, and two-dimensional brick layer packing induce different transport properties and anisotropy[25]. Such detailed molecular stacking manipulation can hardly be achieved in NFAs, partially due to the large size of NFA molecules that retard molecular crystallization, resulting in technical difficulties to obtain single crystals. Chemistry finds its way to solve these difficulties, of maintaining the backbone to secure electronic structure, and changing the aliphatic side chains to handle the details of molecular crystallization and interaction. This capability enables us to look into the details on how solid-state packing of NFA molecules that affect the electronic structure and function, and to depict a rational structure-property relationship.

In this work, we have designed and synthesized three fused-ring NFAs with the same molecular skeleton, but exhibiting different stacking properties by modifying side chains, i.e. IDTT-C6-TIC, IDTT-C8-TIC, and IDTT-C10-TIC (named IDTT-CX-TIC) (Fig. 1). This material family affords the capability to investigate molecular packing behavior with systematic chemical structure modification. The change in side chain length induces fine-control of aliphatic chain interaction, even in the complicated BHJ blends, which opens the broader frame for backbone π–π reorganization, and thus leads to different solid-state properties. By rationally modifying the length of the side chains, the NFA backbone can be manipulated from a strong π–π stacking mode to an intermixed packing mode, and to a non-stacking mode. Different from our current understanding, we find that close atom contacts in a non-stacking mode can enable efficient charge carrier hopping transport through close side atom interactions. The optimized OSCs free from end-group π–π stacking yield a superior PCE of 12.7% with reduce non-radiative recombination loss to that of OSCs rely on classic end-group π–π stacking formed major transport channels, which is not cognized in the organic photovoltaic community. More importantly, molecular and crystal engineering allow us to combine the two solid-state packing motifs together in a BHJ blend, leading to a PCE of 13.7%, surpassing the single-mode interaction dictated function. We believe these observations are vital in new organic semiconductor material design that better performance can be obtained through detailed crystalline structure manipulation.

## Results

**Crystal packing and morphology investigation.** The chemical structures of the three NFAs are shown in Fig. 1. The frontier molecular orbital (FMO) energy levels of IDTT-CX-TIC were measured by cyclic voltammetry (CV) and summarized in Supplementary Fig. 1a and Supplementary Table 1. With increasing the length of the alkyl side chains on IDTT-CX-TIC, the highest

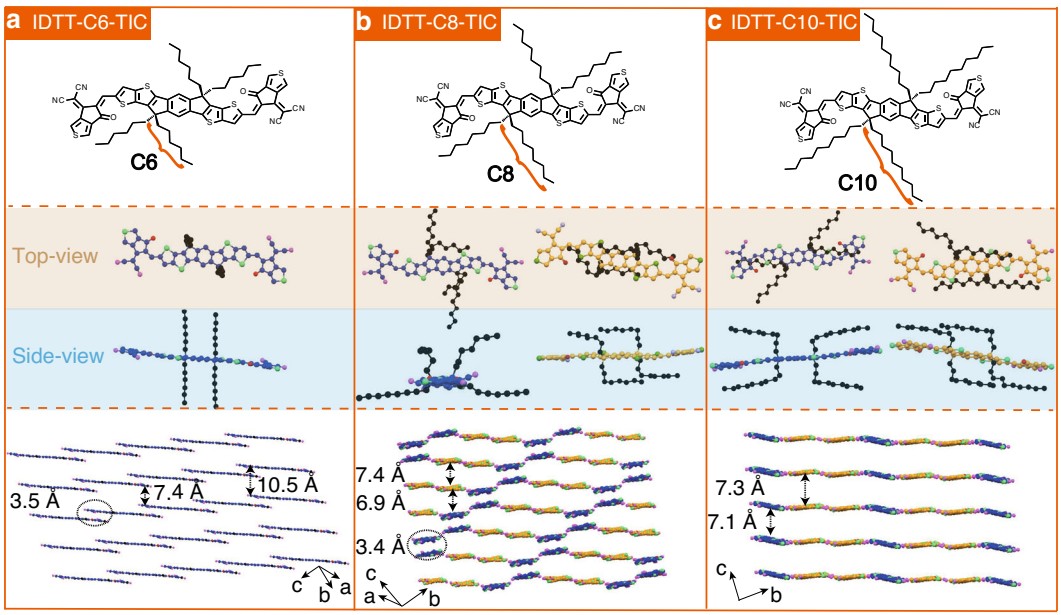

**Fig. 1 Molecular and single crystal structure of IDTT-CX-TIC. a–c** The chemical structure and crystal packing of IDTT-CX-TIC with plane to plane distances. In IDTT-C6-TIC, hexyl side chains take a fully extended geometry that stretching out from the conjugated backbone. In IDTT-C8-TIC, octyl side chains adopt a random extension without structure order and crankshaft-type side chain extension. In IDTT-C10-TIC, octyl side chains adopt harpoon type and crankshaft-type side chain extensions. The molecular backbones with different types of side chain extension are marked by blue and yellow color. C6, C8, and C10 represent n-hexyl, n-octyl, and n-decyl side chains, respectively.

occupied molecular orbital (HOMO) energy levels gradually decrease (IDTT-C6-TIC: $-5.55$ eV; IDTT-C8-TIC: $-5.64$ eV; IDTT-C10-TIC: $-5.71$ eV), and the lowest unoccupied molecular orbital (LUMO) energy levels gradually increase (IDTT-C6-TIC: $-3.99$ eV; IDTT-C8-TIC: $-3.97$ eV; IDTT-C10-TIC: $-3.91$ eV). IDTT-C6-TIC, IDTT-C8-TIC and IDTT-C10-TIC thin films showed similar ultraviolet–visible absorption spectra (Supplementary Fig. 1b) with slightly shifted absorption peak maxima ($\lambda_{max}$) and absorption onsets ($\lambda_{onset}$). Molecule with the medium side chain possesses the lowest optical bandgap ($E_g^{opt} = 1.52$, 1.50 and 1.56 eV for IDTT-C6-TIC, IDTT-C8-TIC and IDTT-C10-TIC, respectively), caused by their different solid-state molecular packing modes, as illustrated in Fig. 1 (the details of IDTT-CX-TIC single molecular structure are shown in Supplementary Figs. 2–4). In IDTT-C6-TIC, hexyl side chains take a fully extended geometry that stretching out from the conjugated backbone. The adjacent molecules in the $\pi$ plane normal direction are connected through side chain interdigitation, as revealed by the close interaction with an average distance of 10.676 Å in Supplementary Fig. 5. The long axis of the molecules forms a close contact through end-group overlapping, with a distance of 3.546 Å (Supplementary Fig. 6), which falls in $\pi$–$\pi$ stacking region that facilitates charge transport. The coupling transfer integral is estimated to be 12.7 meV (Supplementary Fig. 7). The overlapping of end group in a long distance $\pi$ plane normal direction yields a structure feature that one molecule end is inserted in a molecular pair, and the 2D iteration in this manner leads to a network that 2D carrier hopping can be achieved. Such a 2D web then overlays with each other (c-axis direction) to form a 3D molecular registration. It is interesting to note that molecular backbones form slip-stack stacking. CN group is in quite close distance with adjacent ones (~3.1 Å, see Supplementary Fig. 5) that weak electronic coupling exists, 9.2 meV, marking a 3D transportation scenario in the solid state. The simulated carrier mobility of IDTT-C6-TIC is as high as $2.4 \times 10^{-3}$ cm$^2$ V$^{-1}$ s$^{-1}$ (see Supplementary Table 2).

IDTT-C8-TIC molecules show unique solid-state packing as can be tracked from single crystal diffraction. The slightly longer alkyl side chains lead to different molecular geometry in single crystals, as shown in Supplementary Fig. 8, from which we can extract three pairs of interactions, (1) frustrated side chain interaction, (2) symmetric side chain interaction, and (3) mixed side chain interaction. In frustrated side chain model, octyl side chains adopt a random extension without structure order, and thus close backbone packing is achieved. In symmetric side chain model, the long contour length would lead to good aliphatic side chain interdigitation as seen in IDTT-C6-TIC, yet a fully extended geometry would result in reduced $\pi$–$\pi$ interaction, such a balance bends over octyl side chain to form a crankshaft structure. The mixed side chain interaction was found between adjacent molecules of the two pairs. Such a complexity leads to an anomalous solid-state packing in single crystal, which could be dissected into two crystalline domains as marked by blue and yellow color region in Fig. 1b. The frustrated interaction contact yields a transfer integral of 35.2 meV (Supplementary Fig. 9), while the symmetric interaction contact and the mixed interaction model yield extremely low transfer integral of 0.1 and 0.2 meV, respectively. Thus, a highly conductive 1D line is imbedded in a nearly insulating crystalline matrix, making IDTT-C8-TIC unique in NFA materials. There are trivial close atom contacts outside of $\pi$–$\pi$ interaction direction, within partial 0.5 meV range, leading to slow hopping rate for carrier to transport. A low global mobility is expected by integrating all contact types and directions. As a result, the simulated carrier mobility of IDTT-C8-TIC is $2.9 \times 10^{-4}$ cm$^2$ V$^{-1}$ s$^{-1}$ (see Supplementary Table 2), almost one order of magnitude lower than that of IDTT-C6-TIC.

IDTT-C10-TIC in single crystal also takes two different pairs of interactions, as shown in Supplementary Fig. 10, (1) the harpoon type of side chain extension, and (2) the crankshaft type of side chain extension. In both models, a large $\pi$–$\pi$ stacking distance is recorded with a low transfer integral of close to 0 meV (Supplementary Fig. 11). The molecular packing is similar to the crankshaft-packing motif in IDTT-C8-TIC crystals, and a relatively low mobility is expected since the main carrier hopping channel is through close side atom interactions. The simulated carrier mobility is $6.5 \times 10^{-5}$ cm$^2$ V$^{-1}$ s$^{-1}$ (see Supplementary Table 2). Such observation through changing side chain substitution yields quite interesting structure-property relationship that the molecular signature could be transformed into solid states. IDTT-C8-TIC in crystal is an ordered mixture of IDTT-C6-TIC and IDTT-C10-TIC with a slight difference in stacking slipping. We thus could dictate the solid-state structure of NFAs by this simple but elegant strategy to optimize the optoelectronic properties and performance of organic electronic devices.

It is rational that structure order in thin film can be different from that in single crystal due to the non-equilibrium nature of the casting method[20,26–29]. Polymorph and orientation distribution would add more complexity into thin film structure order in grazing incidence wide-angle x-ray diffraction (GIWAXS) results. However, the major molecular packing motif should be translated from single crystal to thin solid films, since the packing mode in single crystal is an energetically favorable state. IDTT-CX-TIC series are quite robust in solid-state packing, and major diffraction peak can be traced from single crystal unit cell with orientation broadening in diffraction peaks (Fig. 2), which are then crosschecked from simulated GIWAXS patterns from specific basal planes. The detailed peak labeling and the related molecular packing motif, for example, the IDTT-C6-TIC (110) crystal plane, the IDTT-C8-TIC (121) crystal plane, and the IDTT-C10-TIC (112) crystal plane, have been presented in Supplementary Fig. 12.

IDTT-C6-TIC presented three diffraction rings at 0.40, 0.60, 0.73 Å$^{-1}$, corresponding to the (001), (010)/(100) and (1-10) miller planes in single crystal, respectively. The 1.8 Å$^{-1}$ diffraction peak corresponds to the $\pi$–$\pi$ stacking, and thus a random orientation is recorded. For IDTT-C8-TIC, more detailed Bragg spots are resolved in small $q$ regions, matching well with the single crystal unit cell. A primary (01$-$1) peak is seen in the in-plane direction, and (121) peak is seen in out-of-plane, indicating a good face-on lamellar order. The (242) plane is seen in 1.61 Å$^{-1}$, and it first-order peak (121) is located in lower $q$ region. There are a series of eyebrow peaks that labeled as (011), (021), (120) and (11-2), ($-$122) miller planes according to single-crystal structure and 2D GIWAXS simulated results. A broad diffraction peak is seen at 1.83 Å$^{-1}$ in the out-of-plane direction, which marks the close $\pi$–$\pi$ stacking in thin film. IDTT-C10-TIC shows a simple diffraction feature with two major rings at 0.85 and 1.70 Å$^{-1}$, which correspond to (112) and (224) miller planes, of the largely spaced $\pi$–$\pi$ stacking distance (7.39 Å). Such a large $\pi$–$\pi$ distance is due to the alkyl chain spacing between adjacent molecules, which is uncorrelated with the charge-transfer process. Other miller planes are also labeled, which, however, are of much lower intensity, indicating the folded alkyl chain crystallization is the major driving force for such a molecular assembly.

The morphology of PBT1-C:IDTT-CX-TIC blends were studied using the combination of GIWAXS, atomic force microscopy (AFM), transmission electron microscopy (TEM), and resonant soft x-ray scattering (RSoXS) methods. As seen in GIWAXS 2D diffraction patterns and line-cut profiles (Fig. 3), the major structure features in NFAs are preserved in the blends (see Supplementary Tables 3–5). These samples show typical polymer (100) diffraction at 0.28 A$^{-1}$, and its crystal coherence length (CCL) is around 9 nm. The (001), (100), and (1-10) diffraction rings are seen in IDTT-C6-TIC neat

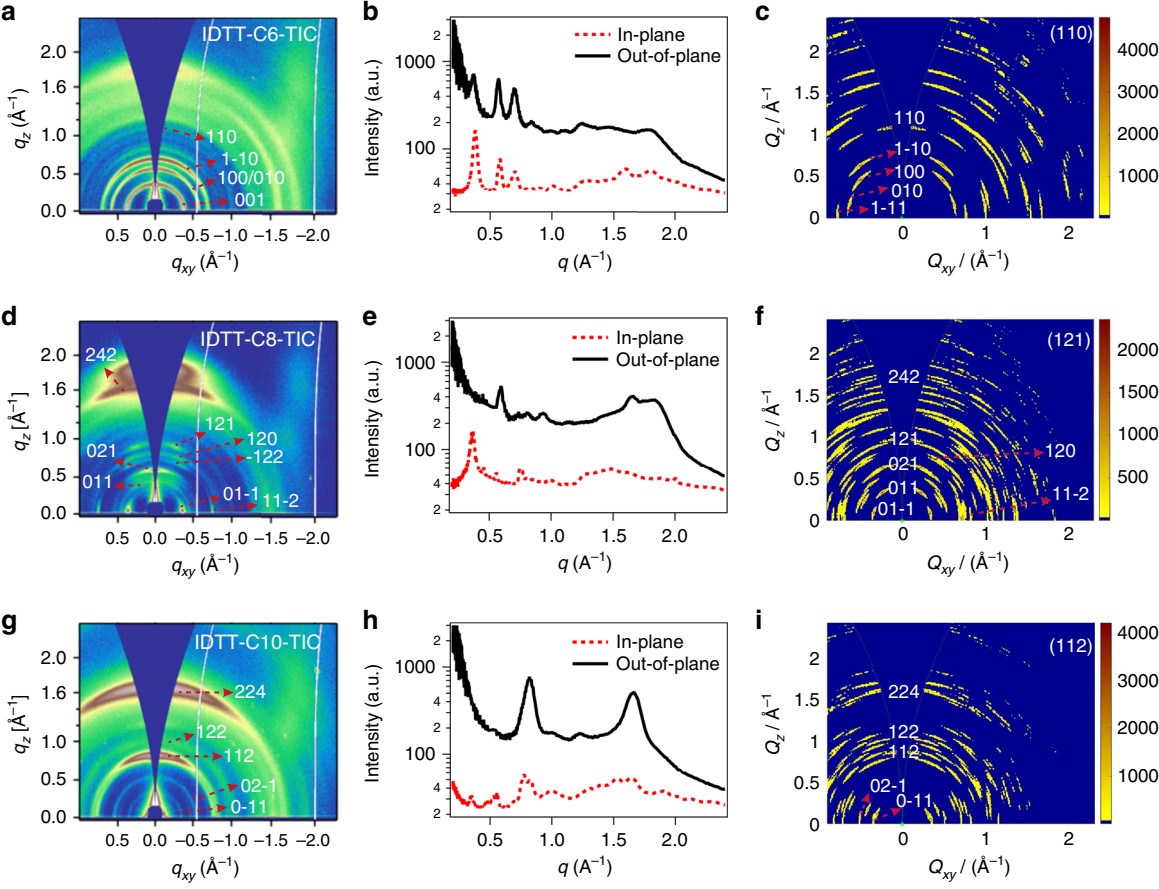

**Fig. 2 GIWAXS and simulated data. a, d, g** GIWAXS diffraction patterns of IDTT-CX-TIC neat films and (**b, e, h**) the corresponding in-plane and out-of-plane line-cut profiles. **c, f, i** The simulated results of IDTT-CX-TIC unit cells. The major molecular packing peaks with the detailed peak labeling was translated from single crystal to thin film. It can be seen that the major diffraction peaks of IDTT-CX-TIC films can be well traced from single-crystal unit cell.

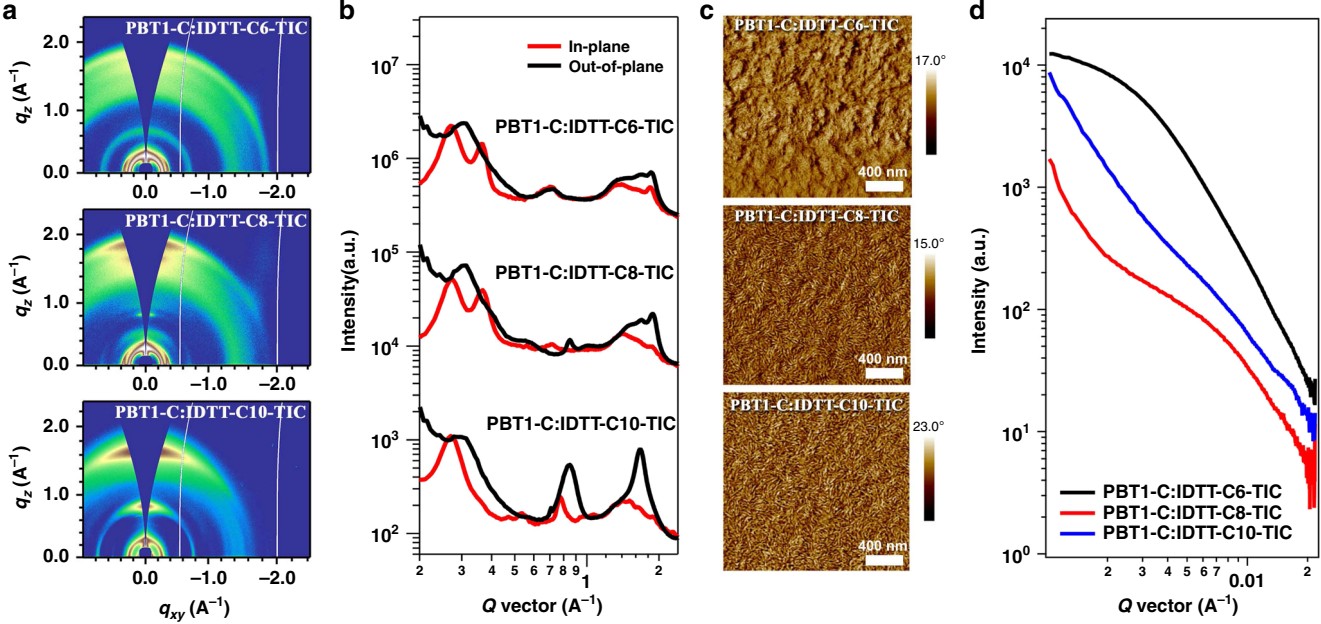

**Fig. 3 Morphology characterizations. a** GIWAXS diffraction patterns of PBT1-C:IDTT-CX-TIC blends and (**b**) the corresponding in-plane and out-of-plane line-cut profiles. **c** AFM phase images of PBT1-C:IDTT-CX-TIC blends (2 × 2 μm). **d** RSoXS scattering profiles of PBT1-C:IDTT-CX-TIC blends.

**Table 1 Summary of device parameters and mobility of PBT1-C:IDTT-CX-TIC solar cells.**

| Active layer | $V_{OC}$ (V) | $J_{SC}$ (mA cm$^{-2}$) | FF (%) | PCE$^a$ (%) | PCE$_{max}$ (%) | $\mu_e$ (cm$^2$ V$^{-1}$ s$^{-1}$) | $\mu_h$ (cm$^2$ V$^{-1}$ s$^{-1}$) |
|---|---|---|---|---|---|---|---|
| PBT1-C:IDTT-C6-TIC | 0.85 ± 0.01 | 17.0 ± 0.3 | 66.7 ± 2.4 | 9.6 ± 0.5 | 10.0 | 2.0 × 10$^{-4}$ | 7.3 × 10$^{-4}$ |
| PBT1-C:IDTT-C8-TIC | 0.88 ± 0.01 | 20.3 ± 0.2 | 74.6 ± 1.1 | 13.4 ± 0.2 | 13.7 | 1.2 × 10$^{-4}$ | 6.5 × 10$^{-4}$ |
| PBT1-C:IDTT-C10-TIC | 0.98 ± 0.01 | 18.1 ± 0.3 | 71.3 ± 1.3 | 12.5 ± 0.4 | 12.7 | 6.7 × 10$^{-5}$ | 3.0 × 10$^{-4}$ |

$^a$The average parameters were calculated from 20 independent cells.

film, indicating a random orientation. In the blend film, the (001) diffraction peak became narrower in the in-plane direction, and the (100) diffraction peak disappeared. The π–π stacking peak showed a quite broad distribution. Thus, a random orientation is still taken in the blend film. The narrowed (001) diffraction indicated that the IDTT-C6-TIC molecules could be tilted through the short axis of the backbone, which is quite different from that in the neat film. The CCLs of IDTT-C6-TIC decreased from 14.3 nm for the neat film to 10.4 nm for the blend film. In IDTT-C8-TIC neat film, the (01-1), (021), (121), (242) crystal planes indicate its face-on orientation. Regarding the blend film, only (01-1) plane can be obviously observed in the in-plane direction. The CCLs of IDTT-C8-TIC neat and blend film are 12.8 nm and 9 nm, respectively. The IDTT-C10-TIC molecules show a face-on orientation in the neat and blend films. We choose the out-of-plane (112) crystal plane to compare the CCL. The CCLs of IDTT-C10-TIC neat film is around 6.8 nm, which is larger than that (4.6 nm) in the blend film.

Overall, the three blends show well-developed polymer and acceptor crystalline features, which shape the phase separation. The phase images were probed by AFM and TEM (Fig. 3c and Supplementary Fig. 13). As seen in PBT1-C:IDTT-C6-TIC blends that large-sized aggregations (hundreds of nanometers) distributed inside fibril networks. A broad scattering was detected in RSoXS, corresponding to a wide distribution of phase separation lengths with the largest domain size of about 200 nm. These large phases will suffer inefficient charge transfer, leading to suppressed short-circuit current ($J_{SC}$) and fill factor (FF) in OSCs. In contrast, the PBT1-C:IDTT-C8-TIC-blend exhibited much refined morphology, and a typical fibril network was observed with appropriate phase separation. A scattering hump in RSoXS confirmed the moderate phase separation about 90 nm, suggesting weak aggregation of NFAs in the blends. The combination of fibril network highways and NFA aggregations balance the hole and electron transport in the blend, leading to enhanced charge extraction as well as improved $J_{SC}$ and FF[30–32]. The PBT1-C:IDTT-C10-TIC-blend showed smooth morphology with fibril network and weak phase separation due to good mixing between materials. The electron mobility of BHJ blend is still as high as 6.7 × 10$^{-5}$ cm$^2$ V$^{-1}$ s$^{-1}$ (see Table 1), comparable to its neat film.

**Photovoltaic performance, $V_{OC}$ loss, and exciton dynamics.** To understand the effect of solid-state molecular packing on the photovoltaic properties of different BHJ thin films, OSCs based on PBT1-C:IDTT-CX-TIC blends were constructed in a device architecture of ITO/ZnO/PBT1-C:IDTT-CX-TIC/MoO₃/Ag. The current density–voltage (J-V) characteristics of OSCs are displayed in Fig. 4a and the detailed parameters are listed in Table 1. The IDTT-C8-TIC based devices showed the best PCE of 13.7% with the highest $J_{SC}$ of 20.3 mA cm$^{-2}$ and the highest FF of 74.6%. IDTT-C6-TIC and IDTT-C10-TIC based devices exhibited PCEs of 10.0% and 12.7% with $J_{SC}$s of 17.0 and 18.1 mA cm$^{-2}$ and FFs of 66.7% and 71.3%, respectively. The $J_{SC}$ values estimated from the external quantum efficiency (EQE) spectra (Fig. 4b) are consistent with those obtained from the J–V characteristics. We found that the open-circuit voltage ($V_{OC}$) of OSCs is influenced by the microstructure of BHJ blends, i.e. the molecular ordering of NFAs.

Quantification of $V_{OC}$ losses for OSCs based on the three NFAs is shown in Fig. 4c–e and summarized in Table 2. The bandgaps of OSCs were determined from the intersection of the EQE edge (EQE$_{edge}$) and the local EQE maximum, as shown in Supplementary Fig. 14[33]. The bandgap determined in this way takes into account the aggregation of NFAs in the blend films and can avoid underestimation of the bandgap of disordered materials using the absorption onset. The IDTT-C10-TIC-blend film has a slightly higher bandgap (1.61 eV) compared to IDTT-C6-TIC (1.60 eV) and IDTT-C8-TIC (1.59 eV), which agrees very well with the absorbance of π–π stacking in the IDTT-C10-TIC-blend film. Ideal $V_{OC}$ estimated according to the Shockley-Queisser limit ($V_{OC, SQ}$)[34] varies within 0.02 eV for OSCs based on the three NFAs. The radiative recombination limit $V_{OC}$, $V_{OC, rad}$, was determined by calculating the saturated current density ($J_{0, rad}$) for recombination from the Fourier-transform photocurrent spectroscopy (FTPS) spectra fitted by electroluminescence (EL) spectra (Supplementary Figs. 15–18) according to the detailed balance theory[35]. The corresponding non-radiative $V_{OC}$ losses, $\Delta V_{OC, nr}$, which is the difference between the $V_{OC, rad}$ and the measured $V_{OC}$ from J–V characteristics ($V_{OC, meas}$), were determined to be 0.33, 0.32, and 0.26 V for IDTT-C6-TIC, IDTT-C8-TIC, and IDTT-C10-TIC-based OSCs, respectively.

The energy of charge-transfer (CT) ($E_{CT}$) states were determined by simultaneously fitting EL and FTPS spectra according to Marcus theory[36] (Supplementary Fig. 18) and summarized in Table 2. It is shown that the three systems have different charge-transfer states energy of 1.37 eV, 1.41 eV, and 1.50 eV for IDTT-C6-TIC, IDTT-C8-TIC, and IDTT-C10-TIC based solar cells, respectively. The differences in $E_{CT}$ largely explained the variations in $V_{OC}$ and non-radiative $V_{OC}$ losses according to energy gap law[37]. The differences in $E_{CT}$ are most closely related to the LUMO energy differences. The LUMO of IDTT-C10-TIC is −3.91 eV, which is 0.08 eV higher than IDTT-C6-TIC (−3.99 eV) and 0.06 eV higher than that of IDTT-C8-TIC (−3.97 eV). It is notable that this work experimentally proved that the changes in different molecular packing and ordering of NFAs in the BHJ blends significantly affected energy level alignment in the D/A interface and hence the CT state energy as well as the non-radiative $V_{OC}$ losses of OSCs. Considering the shifting of the charge-transfer mode from the conventionally end-group π–π stacking (IDTT-C6-TIC) to a mixed transfer mode (IDTT-C8-TIC) and to a π–π stacking free mode (IDTT-C10-TIC), these findings highlight the importance of designing novel molecular semiconductors free from π–π stacking to further reduce the non-radiative recombination losses of OSCs below 0.15 V[38].

Previous efforts to tune charge transport in organic semiconductors have often come at the expense of exciton transport, prompting us to use femtosecond transient absorption spectroscopy (fs-TAS) to investigate whether these materials suffered this compromise[39–41]. In Fig. 4f, the signals from photo-generated charge populations reach their maximum on the same time-scales (~50 ps) as the complete quenching of the excitons in NFAs. These observations agree with our previous study of NFA blends[42–44], and imply the NFA excitons can efficiently transport cross to the acceptor phase domain in the BHJ. In the sub-10 ps region. Our data analysis shows different prompt charge ratios

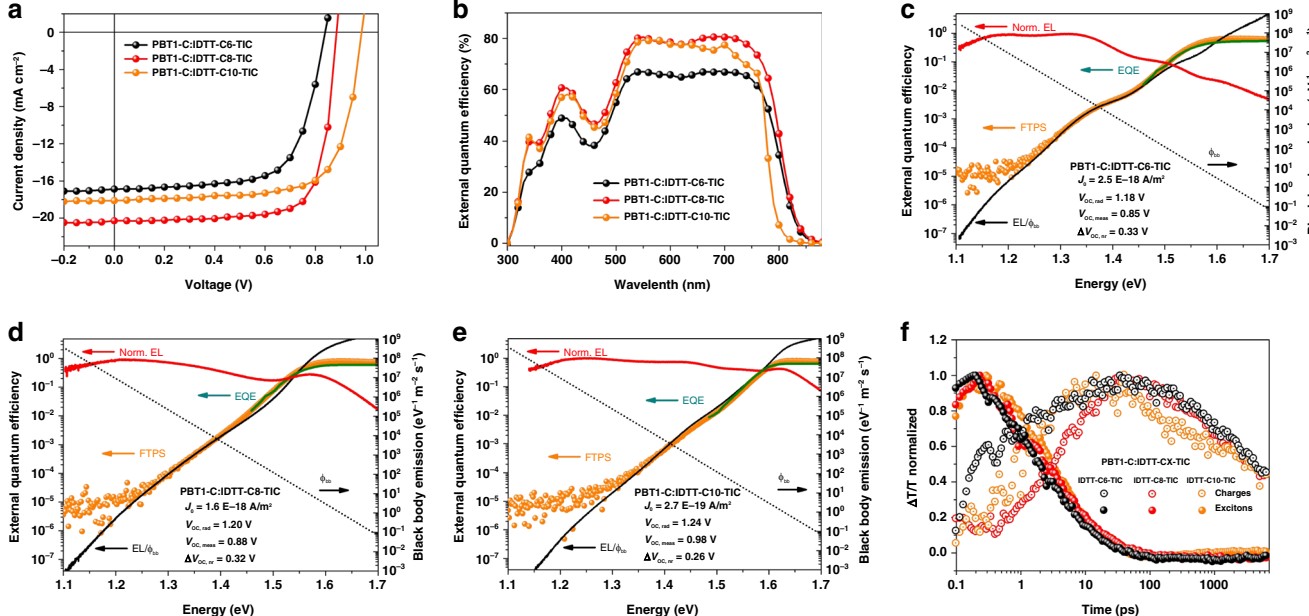

**Fig. 4 Photovoltaic performance, energy loss analysis, and transient absorption kinetics of blend films. a** Current–voltage characteristics of the binary devices based on PBT1-C and IDTT-CX-TIC. **b** External quantum efficiencies (EQEs) of the binary devices. **c, d, e** Semi-logarithmic plots of normalized EL (red line), measured EQE (olive line) and EQE calculated by Fourier-transform photocurrent spectroscopy (FTPS) (orange dots) as a function of energy for devices based on PBT1-C and different acceptors. Dark gray dot line is black body radiation ($\phi_{bb}$) at 300 K. The ratio of EL and $\phi_{bb}$ was used to plot the EQE in the low energy regime (black line). Inset in each figure shows the saturated current density for radiative recombination ($J_0$), radiative $V_{OC}$ limit ($V_{OC, rad}$), measured $V_{OC}$ ($V_{OC, meas}$), and non-radiative $V_{OC}$ losses ($\Delta V_{OC, nr}$). **f** Transient absorption kinetics of excitons and charges in the blends, after the selective excitation of the corresponding acceptor phases at 710 nm at pump fluences of 13 µJ/cm$^2$, 4.7 µJ/cm$^2$ and 4.7 µJ/cm$^2$ respectively. The plots are produced via bilinear decomposition of the TA surfaces using spectral mask of excitons from relevant neat acceptor films (Supplementary Fig. 19).

**Table 2 Measured and calculated parameters to quantify voltage losses.**

| Active layer | $E_g{}^a$ (eV) | $E_{CT}$ (eV) | $J_{0, rad}{}^b$ ($10^{-18}$ A m$^{-2}$) | $V_{OC, SQ}$ (V) | $V_{OC, rad}{}^c$ (V) | $V_{OC, meas}$ (V) | $E_{loss}$ (eV) | $\Delta V_{OC, nr}{}^d$ (V) |
|---|---|---|---|---|---|---|---|---|
| PBT1-C:IDTT-C6-TIC | 1.60 | 1.37 | 2.5 | 1.32 | 1.18 | 0.85 | 0.75 | 0.33 |
| PBT1-C:IDTT-C8-TIC | 1.59 | 1.41 | 1.6 | 1.31 | 1.20 | 0.88 | 0.71 | 0.32 |
| PBT1-C:IDTT-C10-TIC | 1.61 | 1.50 | 0.27 | 1.33 | 1.24 | 0.98 | 0.63 | 0.26 |

$^a E_g$ was determined from the intersection of the absorption edge and the local EQE maximum.
$^b J_{0, rad}$ is the saturated current density for radiative recombination, $J_{0, rad} = q \int_0^\infty EQE(E)\phi_{bb}(E)dE$.
$^c V_{OC, rad}$ is the radiative $V_{OC}$ limit, $V_{OC, rad} = \frac{kT}{q}\ln\left(\frac{J_{sc}}{J_{0,rad}} + 1\right)$.
$^d \Delta V_{OC, nr} = V_{OC, rad} - V_{OC, meas}$.

and charge generation dynamics, however, we are cautious to draw any conclusions from this early time region because charge spectral signatures are buried under very strong exciton signals. We find that shorter exciton lifetimes in the blend compared with the neat NFA films (Supplementary Fig. 20) are consistent with efficient exciton quenching in all cases. Considering the high exciton diffusion coefficients of NFAs, this could still correspond to phase sizes of 10 s of nanometers[45]. We expect that there may be residual unquenched excitons in the IDTT-C6-TIC-blend containing some large phases; however, that signature would be overwhelmed by the majority of excitons undergoing distinctly faster charge generation.

Our finding that exciton transport is rather insensitive to molecular packing configurations—unlike charge transport, and contrary to expectations from earlier generations of organic semiconductors—reflects the photophysical properties of NFAs. Previous models for exciton transport were dominated by hopping between neighbors, and this coupling for exciton transport is not optimized for the same packing configurations as for charge transport[39,40,46]. On the other hand, NFAs are characterized by

strong resonant overlap between their emission and absorption spectra (Supplementary Fig. 1c), resulting in self-Förster radius >4 nm[45]. Thus, we find that differences in local packing structure do not appear to have a significant effect on exciton transport in these materials because of their propensity to undergo through-space exciton hopping over a long distance.

## Discussion

In this work, we demonstrated a simple and effective approach to engineering molecular packing and ordering of NFAs in solid-state BHJ organic photovoltaics. While the recent emergence of advanced NFAs has significantly boosted the PCE of BHJ OSCs by extending the absorption range, improving the charge generation efficiency and minimizing the recombination-related losses, current understanding of charge carrier dynamics and photo-physics is still strongly limited by the empirical model, in which a long-range structural ordering induced by end-group π–π stacking of NFAs is critical for efficient charge transfer and extraction as well as high photovoltaic efficiency. This statement

is particularly true but eliminates the charge-transfer/transport channels induced by close atom interactions, as observed in high-quality organic crystals based on semiconductors free from π–π overlap (Herringbone packing) and systematically investigated in this work. The end-group π–π stacking of NFAs indeed guarantees a high charge carrier mobility and transfer efficiency, however, the ideal long-range structural ordering is difficult to achieve in solid-state thin films, in particular for OSCs based on the BHJ approach. Here we demonstrated that the BHJ thin film free from end-group π–π stacking (IDTT-C10-TIC) can achieve comparable or even superior photovoltaic properties to the one relies on molecular π–π stacking induced transport channels. By rationally modifying the length of side chains, the backbone of NFAs can be manipulated from a strong π–π stacking mode to an intermixed of π–π and no π–π packing mode to refine its solid-state properties.

As it is almost impossible so far to reduce the structural disorder of BHJ blends, the diverse charge transport channels induced by close side atom interactions and π–π stacking could be a promising alternative approach enabling sufficient charge-transfer efficiency in highly disordered material systems. As a result, the optimized OSCs based on the molecules with intermixed packing mode yielded higher PCEs with high $V_{OC}$s, and reduced non-radiative recombination losses than those in devices based on molecules with the classic end-group π–π stacking mode. Therefore, the findings demonstrated in this work not only provide new insights into the effect of NFA molecular packing on exciton dissociation, charge transport, and recombination losses, but also open a new avenue in materials design that endows efficient multiple charge transport channels for next-generation organic photovoltaics.

## Methods

**Materials**. The polymer donor PBT1-C was synthesized via referencing the reported literature[31]. The detailed synthetic procedures of IDTT-CX-TIC can be found in Supplementary Fig. 21, the corresponding NMR data are included as Supplementary Figs. 22–39 and crystal information of IDTT-CX-TIC is listed in Supplementary Table 6.

**Theoretical calculation**. The details of simulations can be found in Supplementary Note 1. The IDTT-CX-TIC crystal based on a supercell ($6 \times 6 \times 4$ for IDTT-C6-TIC; $4 \times 4 \times 3$ for IDTT-C8-TIC; $6 \times 3 \times 3$ for IDTT-C10-TIC) was equilibrated at 300 K for 5 ns (see Supplementary Fig. 40).

**Cyclic voltammetry measurement**. Cyclic voltammetry (CV) measurements were performed on a CHI660E electrochemical workstation in a three-electrode cell in anhydrous acetonitrile solvents solution of $Bu_4NPF_6$ (0.1 M) with a scan rate of 50 mV/s at room temperature under argon. A $Ag/Ag^+$ wire, two platinum wires were used as the reference electrode, counter electrode, and working electrode, respectively. The materials to be tested in chloroform solution were dried on the surface of the working electrode. The potential of $Ag/Ag^+$ reference electrode was internally calibrated by using ferrocene/ ferrocenium ($Fc/Fc^+$) as the redox couple.

**Ultraviolet–visible absorption**. Ultraviolet–visible absorption spectra were acquired on a UV–vis spectrophotometer (Shimadzu UV-2700). All film samples were spin cast on quartz glass substrates.

**TEM**. TEM studies were conducted with a JEOL JEM-1400 microscope. The samples for electron microscopy were prepared by dissolving the PEDOT:PSS layer using water and transferring the floating active layer to the TEM grids.

**AFM**. Atomic force microscopy images were investigated on a Dimension Icon AFM (Bruker) in a tapping mode. All film samples were spin cast on indium tin oxide (ITO)/ZnO substrates.

**GIWAXS**. The GIWAXS characterization of the thin films is measured on beamline 7.3.3 at the Advanced Light Source (Lawrence Berkeley National Laboratory). All samples are prepared under device conditions on the silicon wafer substrate. The scattering signal of samples is recorded with a pixel size of 0.172 mm by 0.172 mm (Pilatus 2 M detector). The distance between the samples and beam

center is ≈ 300 mm which calibrated by the silver behenate standard. The incidence angle is set to be 0.16°. The beam energy is 10 keV, operating in top-off mode. A 30 s exposure time on a 2D charge-coupled device (CCD) detector is recorded to collect the diffraction signals. All GIWAXS measurements are done in a helium atmosphere.

**RSoXS**. The RSoXS is measured at beamline 11.0.1.2 at Advanced Light Source, Lawrence Berkeley National Laboratory. All samples are prepared under device conditions on the Si/PEDOT:PSS substrates. The blend films are then floated in water and transferred to a silicon nitride window. The scattering signals are collected in a vacuum by using a Princeton Instrument PI-MTE CCD (charge-coupled device) camera.

**EL and FTPS measurements**. Electroluminescence measurements were performed by applying an external voltage/current source through the devices. The luminescence spectra were collected in a back-scattering geometry, dispersed by an iHR320 monochromator (Horiba Jobin-Yvon) and recorded with a Peltier-cooled Si CCD (Synapse, Horiba Jobin-Yvon). The FTPS measurements were carried out using a Bruker Vertex 70 Fourier-transform infrared (FTIR) spectrometer, equipped with a quartz tungsten halogen lamp and a quartz beamsplitter as well as an external detector option. A low-noise current amplifier (Femto DLPCA-200) was used to amplify the photocurrent produced on illumination of the photovoltaic devices with light modulated by the FTIR. The output voltage of the current amplifier was fed back into the external detector port of the FTIR. Absolute EQE photovoltaic values were redrawn by correcting the FTPS to the EQE of the corresponding solar cells.

**Transient absorption measurements**. The charge photogeneration and recombination dynamics of polymer: fullerene systems were studied using Femtosecond Transient Absorption (TA) spectroscopy, where the system used is according to the design described in previous literature[47,48]. Here all the polymer:fullerene blends under vacuum were excited by 100 fs, 580 nm laser pulses, produced by an optical parametric amplifier (TOPAS) which are chopped at 1.5 KHz. The broadband probe pulses are generated by focusing a small fraction of the 800 nm fundamental into an undoped YAG crystal. The polarization of the pump beam is set at the magic angle (54.70) with respect to that of the probe beam in order to avoid the orientational or polarization effects. After the transmission through the sample, the probe continuum is spectrally dispersed and collected (visible and near IR components) simultaneously by two cameras. The differential transmission signal is calculated from the transmitted probe pulses corresponding to the pump on versus off. A retroreflector on a motorized translational stage is used to vary the relative delay between the pump and the probe pulses. The result of a typical experiment is a three-dimensional data set; where each column corresponds to a full spectrum at a certain time and each raw corresponds to the kinetics at different wavelengths. Exciton dissociation and charge generation dynamics in NFA blends were produced by multivariate curve resolution by alternating least square (MCR-ALS) method, discussed elsewhere[49,50]. The method includes the bilinear decomposition of the blend TA surfaces by applying spectral masks of excitons from the pristine acceptor films (Supplementary Fig. 19).

**Device development and testing**. The devices were fabricated with an architecture of ITO/ZnO/active layer/$MoO_3$/Ag. The ITO-coated glass substrates were cleaned with sequential ultrasonication in a soap–deionized water mixture, deionized water, acetone, and isopropanol. The washed substrates were dried at 110 °C for one night. The ZnO precursor solution was spin coated on the ITO substrates at 4000 rpm and the ZnO layer was generated at 200 °C for 15 min in the ambient atmosphere. The substrates were then transferred into a nitrogen-filled glove box. Subsequently, the active layer was spin coated on the ZnO layer via spin-coating from a chloroform solution of PBT1-C:IDTT-CX-TIC. The $MoO_3$ (3 nm) and Ag electrode (100 nm) were deposited by the sequential thermal evaporation. The current density–voltage ($J$–$V$) characteristics of the PV devices were measured under $N_2$ conditions using a Keithley 2400 Source Measure Unit. The currents were measured under 100 mW cm$^{-2}$ simulated 1.5 Global (AM 1.5 G) solar simulator (Enli Technology Co., Ltd, SS-F5-3A). The light intensity was calibrated by a standard Si solar cell (SRC-2020, Enli Technology Co., Ltd). The $J$–$V$ curves were measured along the forward scan direction from −0.5 to 1.1 V with a scan step of 50 mV and a dwell time is 10 ms. EQE spectra were carried out using a solar-cell spectral-response measurement system (QE-R, Enlitech). The current density–voltage ($J$–$V$) curve of representative devices with various PBT1-C:IDTT-CX-TIC ratio and additive volume are shown in Supplementary Figs. 41–46, and the corresponding photovoltaic parameters are listed in Supplementary Tables 7 and 8. We have performed stability analysis (Supplementary Fig. 47)

**SCLC mobility measurements**. SCLCs were tested in electron-only devices configured with the ITO/ZnO/active layer/ZrAcac/Al and hole-only devices configured with the ITO/PEDOT:PSS/active layer/$MoO_3$/Ag. The current density–voltage ($J$–$V$) characteristics of the hole or electron-only devices are fitted by the

Mott–Gurney law:

$$J = (9/8)\varepsilon_r\varepsilon_0\mu(V^2/L^3),$$

where $J$ is the current density, $\varepsilon_r$ is the dielectric permittivity of the active layer, $\varepsilon_0$ is the vacuum permittivity, $L$ is the thickness of the active layer, $\mu$ is the mobility. $V = V_{app} - V_{bi}$, where $V_{app}$ is the applied voltage and $V_{bi}$ is the offset voltage ($V_{bi}$ is 0 V here). The mobility can be calculated from the slope of the $J^{0.5}{\sim}V$ curves. Characteristic curves of SCLC devices are shown in Supplementary Fig. 48.

**Reporting summary**. Further information on research design is available in the Nature Research Reporting Summary linked to this paper.

## Data availability

The data supporting the results of this work are available from the corresponding authors upon reasonable request. The X-ray crystallographic coordinates for structures reported in this study have been deposited at the Cambridge Crystallographic Data Centre (CCDC), under deposition numbers 1974016-1974018. These data can be obtained free of charge from The Cambridge Crystallographic Data Centre via www.ccdc.cam.ac.uk/data_request/cif.

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

## Acknowledgements

This work was financially supported by the National Natural Science Foundation of China (NSFC) (Grant Nos. 51825301, 21734001, 51973110, 21734009, 21733005, and 51761135101) and the 111 Project (Grant B14009). F.L. gratefully acknowledges the support by Beijing National Laboratory for Molecular Sciences (BNLMS201902). N.L. gratefully acknowledges the support of 111 project (D18023) and the DFG research grant: BR 4031/13-1. C.J.B. gratefully acknowledges the financial support through the "Aufbruch Bayern" initiative of the state of Bavaria (EnCN and SFF), the Bavarian Initiative "Solar Technologies go Hybrid" (SolTech), and DFG SFB953 (project no. 182849149) and DFG INST 90/917-1 FUGG. Y.S. gratefully acknowledges Prof. Erjun Zhou and Dr. Ailing Tang (NCNST) for the assistance with SCLC measurements. X-ray data were acquired at beam lines 7.3.3 and 11.0.1.2 at the Advanced Light Source, Lawrence Berkeley National Laboratory, which is supported by the Director, Office of Science, Office of Basic Energy Sciences, of the U.S. Department of Energy under Contract No. DE-AC02-05CH11231.

## Author contributions

L.Y. synthesized and characterized the IDTT-CX-TIC. K.W. fabricated and characterized the devices. J.X. and F.L. performed the morphology characterization and analysis. X.D., N.L. and C.J.B. studied the energy loss of OSCs. S.C., K.C. and J.M.H. measured TAS and performed the analysis. J.Z., and Z.X. grew IDTT-CX-TIC single crystals. S.T. participated in the discussion of IDTT-CX-TIC synthesis. G.H., and Y.Y. calculated the electronic coupling of IDTT-CX-TIC. N.L., F.L., and Y.S. supervised and directed this project. L.Y., K.W., X.D., N.L., F.L., and Y.S. wrote the manuscript. All authors commented on the manuscript.

## Competing interests

The authors declare no competing interests.
