## [Peer Review File · Nature Communications]

Reviewers' comments:

Reviewer #1 (Remarks to the Author):

The authors reported that side-chain engineering of non-fullerene acceptors (NFAs) could drive the fused-ring backbone assembly from a π - π stacking mode to an intermixed packing mode, and to a non-stacking mode. Despite no obvious π - π stacking existed in IDTT-C10-TIC, it was surprisingly found that high efficiency and small non-radiative recombination loss could be achieved in its corresponding devices. Moreover, the intermixed solid-state packing motif (π - π / no π - π stacking) in active layers could enable OSCs with the highest efficiency, surpassing the single mode interaction dictated function. The reported results are interesting and somehow different from our current understanding of the role of π - π stacking in influencing the long-range structure ordering of fused-ring NFAs. This work can provide new insights into designing high-performance NFAs. I would recommend the publication of this paper in Nat. Commun. with minor revisions.

1. The analysis of single crystal structure and then extended to thin film structure is interesting. The authors should comment more on how to make this move. It is seen in thin film GIXD that major diffraction peaks show quite strong azimuthal angle distribution, which should be commented.

2. As shown in Figure 2g and f, IDTT-C10-TIC showed a strong diffraction peak at 1.60 \AA^{-1} , which typically refers to the π - π stacking. However, here, the authors claimed that this diffraction peak corresponds to (202) miller plane. The authors should interpret it clear. In addition, how about the molecular packing of IDTT-C10-TIC relative to the substrate and why the signal of the diffraction peak at 1.60 \AA^{-1} is so strong?

3. It is quite interesting to see the change of device Voc with respect to different NFA molecules. Can this observation be correlated with molecular structure and solid-state structure?

4. As shown in Table 2, IDTT-C10-TIC-based devices exhibited the lowest non-radiative recombination. What is the underlying reason? Why NFAs free from π - π stacking can lead to reduced non-radiative recombination losses in OSCs?

5. The authors used PBT1-C as the polymer donor to blend with IDTT-CX-TIC. Have the authors tried other polymer donors (e.g. PM6) as the selection of appropriate donor materials is crucially important in realizing high-efficiency OSCs.

Reviewer #2 (Remarks to the Author):

In this work, by increasing the length of the linear sidechains attached to the same NFA acceptor molecular skeleton, the molecular interaction is tuned from the typical end-group π - π stacking to a non-stacking packing motive. These significant changes in packing and intermolecular coupling go along with distinct changes of the electrical, electronic and photovoltaic properties. Most importantly, the authors find a continuous increase of the Voc (and a decrease of the non-radiative loss) upon suppression of the π - π interaction while on the other hand, losses in FF due to a reduced carrier mobility are rather small. It is concluded that these findings are of importance for the future design of molecular semiconductors with reduced non-radiative Voc losses.

The large increase in Voc, related to suppressed π - π interaction is, indeed, remarkable. Nevertheless, the paper falls short of being suited for publication in Nature Communication for several reasons.

Most importantly, while the paper presents a detailed study of the molecular arrangement of the three different NFAs in the neat phases, the Voc is related to the CT state whose properties are determined by the details of the DA interface. There have been several studies on the interplay between molecular arrangement and orientation on the Voc and on the non-radiative losses. For example, the McGehee group studied the role of the intermolecular arrangement on the photovoltaic performance and specially the CT properties C60-based DA systems (e.g.

10.1021/ja502985g, 10.1002/adma.201301319). Also the relative orientation of the donor with respect to the DA interface has been shown to affect the CT energy and radiative losses, which was assigned to different vibronic coupling (10.1038/s41467-017-00107-4, 10.1002/aenm.201601325). Despite the importance of these effects, information on the DA molecular interactions is almost absent in the present submission. The description of the morphologies of the blend layers is rather short and there is no discussion of e.g. the relative orientation of the molecules in the donor and acceptor phases etc.

My second criticism concerns the analysis of the EQE and EL spectra with regard to the CT properties. It is known from other systems that the absorption and emission properties of the NFA singlets become important in systems with a small driving force (10.1021/jacs.9b01465). In fact, what the authors describe as multiply charge transfer states may be a superposition of NFA exciton and CT state absorption and emission. That the different peaks in the EL spectra have different origins is also suggested by the bias dependent EL spectra in Figure S16. Therefore, a proper decomposition of the spectra in the contributions from CT and singlet states is mandatory. This is usually done by recording the EQE and EL of the neat NFA layers. With such a proper deconvolution, an exact determination of the CT energy will be feasible.

Finally, a link between the different crystal structures and the CT (Voc) properties is missing. Is the increase of the CT energy with increasing side chain length simply related to an increase in the LUMO energy of the NDA (which would be a trivial and well-known fact)? Or are there additional factors (relative orientation etc.) contributing to the interplay between acceptor molecular packing and CT energy. Here, I remind the authors that the CT energy and decay kinetics is primarily determined by the properties of the DA interface, and that there are numerous mechanisms how these properties are affected by the packing and orientation of the molecules in the individual phases.

Minor points:

The captions of the Figures in the SI are short and often do not provide sufficient information. E.g. it is not stated whether the crystal structures in Figures 5-11 are from X-ray diffraction experiments on singly crystals, from MD simulations, or from a combination of both.

Figure S1 and Table S1: In my opinion, the values of the LUMO in Table S1 do not correspond to the CV curves in Figure S1. For example, IDTT-C8-TIC displays the most negative onset of the redox process, which is in contrast to the values in Table S1. I was also surprised of the fact that the reduction is not reversible for any of the three NFAs (which makes an exact determination of the LUMO energy nearly impossible). A detailed discussion of the JCV curves and their analysis is needed.

The directions of the crystallographic axes should be added to Figure S5-11 for a better understanding of the molecular packing.

Figure S46 and Table S2. According to the methods section, the current density–voltage (J–V) characteristics of the hole or electron only devices were fitted by the Mott–Gurney law. According to this law, the current increases strictly quadratically with voltage, meaning that the slope of the JV curves (in the log-log plots in Figure S36) should all have the same slope of 2. This is clearly not the case. A detailed discussion of the JV curves and their analysis is needed.

In my opinion, the TAS data are superfluous as they do not contribute to the understanding of the interplay between molecular packing and the CT properties. Moreover, the results from TAS (where all three systems display similar exciton lifetimes and similar charge formation kinetics) are in contradiction to the structural data (Figure 3 and corresponding text), where it is stated that the large scale phase separation of the PBT1-C:IDTT-C6-TIC blend “is detrimental to efficient charge transfer”.

Reviewer #3 (Remarks to the Author):

In this work, the authors rationally designed non-fullerene acceptor (NFA) molecules by changing the side chain length to realize different molecular stacking modes and their impact on charge transfer dynamics, non-radiative recombination loss and overall device performance.

The authors systematically studied molecular structures of NFAs and modelled their potential stacking modes. They showed that end group-facilitated stacking and long-range order can be replaced with an intermixed stacking or even a non-stacking mode, with latter being overlooked by the community, to achieve superior performance.

Although there is some new understanding obtained through this work regarding to molecular design endeavor in the future, universality of this approach is yet to be demonstrated for other and more efficient (Y-series) NFAs, and novelty of investigating the influence of molecular engineering on photovoltaic performance is somewhat arguable here. I also believe some of the findings, such as the reason for the difference in photovoltaic performance (charge generation, charge transport etc.) are not backed conclusively. Overall, side chain engineering is not anymore considered as a novel strategy and I do not find the manuscript suitable for Nature Comm. caliber unfortunately.

In general, the article is well written with an elaborate language and few minor mistakes (typos, wrong wording, repetition of words with similar meanings etc.) to be corrected here and there (also in Supplementary Information). I would also suggest the use of shorter sentences in some parts for a better read.

Comment 1: Power conversion efficiencies (PCEs) reported for NFA organic solar cells are now above 17% (>18% in Science Bulletin). It would be better, if authors update the information on Page 4.

Comment 2: It is not clear to me whether molecular stacking manipulation in NFAs is challenging. Initially, it is indicated that obtaining single crystals is difficult. Then, in the very next sentence, it is said chemistry can solve such problems which is regarded as molecular engineering and widely used by scientists to synthesize analogues of archetypal NFAs such as Y6 for various improvements in organic solar cell (OSC) performance.

Comment 3: It would be a better guide to the eye, if the backbone, end groups and side chains of molecules were in different colors in Sup Fig 8 and 10 like presented in Sup Fig 3. It is hard to follow what is depicted in the figures in their current states. Also, incorporating the interaction types observed in IDTT-C8-TIC and IDTT-C10-TIC into respective figures in SI would make it easier to follow.

Comment 4: Is there a particular rationale behind using IT core for the molecules reported in this study rather than electron-deficient core adopted in Y-series non-fullerene acceptors which outperform this class of NFAs? Is this alkyl chain-dependent packing mode engineering a universally applicable approach for other NFAs?

Comment 5: Crystal packing and morphology investigation section could use few references regarding to assessment of π - π stacking distances, how alkyl chains impact charge transfer process and their crystallization the molecular assembly, and estimation of crystal coherence lengths on Page 10 and 11.

Comment 6: The electron mobility of PBT1-C:IDTT-C10-TIC is on the order of 10^{-4} in Table 1, in contrast to what is stated on Page 13 (on the order of 10^{-5}). Also, based on Table 1 and Supplementary Table 2, electron mobility of the blend is higher than that of neat acceptor film which is surprising.

Comment 7: Could authors expand on the influence of the molecular ordering of NFAs on VOCs? How exactly does the microstructure of BHJ blends govern the VOC?

Comment 8: I am wondering how Eloss is calculated in units of V, where the unit of E_g is eV in Table 2. One way to correct this would be multiplying each type of VOC with q to convert the units to eV.

Comment 9: Do authors have any comments on how PBT1-C:IDTT-C6-TIC blend system exhibits competitive, if not superior, electron and hole mobilities despite its large aggregates and large-scale phase separation compared to other two blend systems? If this is an expected outcome, why are the JSC and FF significantly lower? According to the findings of this study, exciton diffusion is

not drastically affected by the crystal structures and molecular packings either. The work cited at the end of Results section mainly investigates the IDIC molecule, whereas this study looks into 3 different molecules purposefully engineered to observe the effect of side chain length on molecular packing and eventually, non-radiative VOC loss. Thus, it would be more conclusive if there was direct evidence of low energetic disorder and self-spectral overlap for respective systems presented here.

Comment 10: I would suggest an edit for the sentence on Page 14 as "... the lowest CT state energy increases with increasing the side chain length, ...". In its current version of "... the lowest CT states energy in each system is increased with increasing the length of side chain, ...", it is read like each blend system has also a variation of side chain length in itself, in fact the side chain length is what separates each blend system.

Comment 11: Although the non-radiative VOC losses were finetuned by side chain length engineering, the BHJ blend with the lowest non-radiative VOC loss is not the most efficient solar cell in this study. Hence, I am not sure how sound the message that is tried to get across to by the authors at the bottom of Page 14 and top of Page 15. Evidently, reducing the non-radiative VOC loss by designing molecules free from n-n stacking appears promising, if it also leads to higher efficiencies compared to all the other charge transfer modes.

Reviewers' comments:

Reviewer #1 (Remarks to the Author):

The authors reported that side-chain engineering of non-fullerene acceptors (NFAs) could drive the fused-ring backbone assembly from a π - π stacking mode to an intermixed packing mode, and to a non-stacking mode. Despite no obvious π - π stacking existed in IDTT-C10-TIC, it was surprisingly found that high efficiency and small non-radiative recombination loss could be achieved in its corresponding devices. Moreover, the intermixed solid-state packing motif (π - π / no π - π stacking) in active layers could enable OSCs with the highest efficiency, surpassing the single mode interaction dictated function. The reported results are interesting and somehow different from our current understanding of the role of π - π stacking in influencing the long-range structure ordering of fused-ring NFAs. This work can provide new insights into designing high-performance NFAs. I would recommend the publication of this paper in Nat. Commun. with minor revisions.

Answer: We greatly thank the reviewer for the very positive evaluation of our work.

1. The analysis of single crystal structure and then extended to thin film structure is interesting. The authors should comment more on how to make this move. It is seen in thin film GIXD that major diffraction peaks show quite strong azimuthal angle distribution, which should be commented.

Answer: The general thinking behind is that a specific material should have a favorable solid-state packing, excluding the polymorph effect. We take the single crystal structure as the stable form, and try to connect it with thin film morphology, which consists of multiple crystals with different orientations. We take different basal plane, and run GIWAXS simulation to find out the overall orientation of the single film crystallites. As the reviewer commented, these crystallites show strong azimuthal angle distribution, which is due to the surface heterogeneous nucleation effect. However, there is no uniform driving force that directs the overall crystal growth, which is quite common in the formation of organic semiconductor thin films.

2. As shown in Figure 2g and f, IDTT-C10-TIC showed a strong diffraction peak at 1.60 \AA^{-1} , which typically refers to the π - π stacking. However, here, the authors claimed that this diffraction peak corresponds to (202) miller plane. The authors should interpret it clear. In addition, how about the molecular packing of IDTT-C10-TIC relative to the substrate and why the signal of the diffraction peak at 1.60 \AA^{-1} is so strong?

Answer: The diffraction peak allocation should be traced back to the molecular stacking of a stable form. We obtained the single crystal structure, from which we could identify the details of molecular interactions. We double checked the crystal structure and run GIWAXS simulation. New packing structure and peak labeling was added. The peak information was updated and the 1.60 \AA^{-1} is ascribed to the (224) miller plane. We change the peak labeling mainly because (112) miller index series give a physical meaning of molecular packing, which is more accurate comparing to the (202) labeling, though they are of very similar diffraction angles. The 2D GIWAXS diffraction pattern also confirms the conclusion, we find (112) and (224) peak series as analyzed from peak position, full-width-at-half-maxima etc. The molecule take a face-on orientation relative to the substrate. The diffraction peak at 1.60 \AA^{-1} is strong mainly due to good ordering of this plane. As depicted in supporting information, Fig. 12, we noticed aliphatic chain folded crystallization, which helps identify the reason for its strong ordering effect since formation energy in this direction is largely dissipated.

3. It is quite interesting to see the change of device V_{oc} with respect to different NFA molecules. Can this observation be correlated with molecular structure and solid-state structure?

Answer: The NFAs are developed based on the same backbone, thus the major factor that induces V_{oc} change should be correlated with different packing motif of NFAs. It should also be noted that V_{oc} is not only materials dependent, but also determined by the device configuration and others. At the current stage we cannot exclusively ascribe the V_{oc} change to a specific factor. Nevertheless, as the solar cells were fabricated under the same conditions in the same device configuration, the solid state molecular packing could be the major factor

that contributes to the V_{oc} change. It is known that strong solid state packing with intermolecular interaction could reduce the bandgap of organic thin films and thus device V_{oc} . In our current work, the packing motif without obvious π - π stacking (IDTT-C10-TIC) leads to a high V_{oc} . Although the correlation is well established, we cannot provide an analytical reason from electronic structure perspective due to the theoretical difficulties.

4. As shown in Table 2, IDTT-C10-TIC-based devices exhibited the lowest non-radiative recombination. What is the underlying reason? Why NFAs free from π - π stacking can lead to reduced non-radiative recombination losses in OSCs?

Answer: Non-radiative losses are inversely dependent on interface charge transfer state energy according to energy gap law (K. Vandewal *et al.* Nat Energy 2017, 2, 17053). In the revised manuscript, we determined the energy of charge transfer state (in the revised Table 2) by fitting EL and FTPS according to Marcus theory (K. Vandewal, Annu. Rev. Phys. Chem. 2016, 67:113-133) (detailed in the revised supporting information Figure S18). We found that the charge transfer state energy is the highest in the IDTT-C10-TIC-based devices, which is one of the main reasons that the non-radiative recombination losses are lower in this system. As for the variations in charge transfer state energy for the three different material systems, we found good correlation of the CT state energy with LUMO energy differences of the acceptors, which is influenced by the molecular packing.

5. The authors used PBT1-C as the polymer donor to blend with IDTT-CX-TIC. Have the authors tried other polymer donors (e.g. PM6) as the selection of appropriate donor materials is crucially important in realizing high-efficiency OSCs.

Answer: As we mainly focus on investigating the effects of solid-state packing on the electronic structure and function of NFA molecules rather than achieving high efficiency in this work, we have not tried other polymer donors. However, due to the current pandemic situation, we have very limited access to our labs to perform required experiments. Nevertheless, we are planning to try more polymer donors in the near future and will report the follow-up results.

Reviewer #2 (Remarks to the Author):

In this work, by increasing the length of the linear sidechains attached to the same NFA acceptor molecular skeleton, the molecular interaction is tuned from the typical end-group pi-pi stacking to a non-stacking packing motive. These significant changes in packing and intermolecular coupling go along with distinct changes of the electrical, electronic and photovoltaic properties. Most importantly, the authors find a continuous increase of the V_{oc} (and a decrease of the non-radiative loss) upon suppression of the pi-pi interaction while on the other hand, losses in FF due to a reduced carrier mobility are rather small. It is concluded that these findings are of importance for the future design of molecular semiconductors with reduced non-radiative V_{oc} losses.

The large increase in V_{oc} , related to suppressed pi-pi interaction is, indeed, remarkable. Nevertheless, the paper falls short of being suited for publication in Nature Communication for several reasons.

1. Most importantly, while the paper presents a detailed study of the molecular arrangement of the three different NFAs in the neat phases, the V_{oc} is related to the CT state whose properties are determined by the details of the DA interface. There have been several studies on the interplay between molecular arrangement and orientation on the V_{oc} and on the non-radiative losses. For example, the McGehee group studied on the role of the intermolecular arrangement on the photovoltaic performance and specially the CT properties C60-based DA systems (e.g. 10.1021/ja502985g, 10.1002/adma.201301319). Also the relative orientation of the donor with respect to the DA interface has been shown to affect the CT energy and radiative losses, which was assigned to different vibronic coupling (10.1038/s41467-017-00107-4, 10.1002/aenm.201601325). Despite the importance of these effects, information on the DA molecular interactions is almost absent in the present submission. The description of the morphologies of the blend layers is rather short and there is no discussion of e.g. the relative orientation of the molecules in the donor and acceptor phases etc.

Answer: We highly appreciated the valuable comments from the reviewer. It is a quite interesting question regarding the DA interface and V_{oc} . A detailed analysis will be quite useful but at the current stage, we could not think of any method that can probe such feature at device scaled level. We thank the reviewer very much for providing the mentioned

references. However, they did not experimentally tackle the D/A interface issue. For example, 10.1038/s41467-017-00107-4 and 10.1002/aenm.201601325 looked into the collective orientation of molecules and their effect, which is not the direct probing of D/A interfaces. Most D/A interaction investigation up to date is based on theoretical models, which yields large amount of information. We are not aware of any report that could carry such research to experimental level. The morphology characterization is limited at the microscopic level and could not provide detailed interfacial information. In fact, the D/A interface and interaction could only be seen as amorphous mixture by using the scattering technique. It is not possible to resolve their relative orientation and interactions. We thought of soft x-ray methods, which however uses much large photons and thus could not be useful to reveal such localized results. Spectroscopic methods such as NEXAFS could only provide collective orientation factor but never any detailed specific interaction modes. We thought of STEM, which however changes the sample preparation condition and thus will be very different from the original purpose. To answer this question, it is required to rely on new techniques emerging in the future. We thus in this report focus on the interesting observation and not discuss on baseless D/A interactions. We noticed that a new packing motif without π - π stacking gives rises of low non-radiative recombination and high V_{oc} , and think this observation is of great significance to the community. That's all we could make a solid conclusion and we thus follow.

2. My second criticism concerns the analysis of the EQE and EL spectra with regard to the CT properties. It is known from other systems that the absorption and emission properties of the NFA singlets become important in systems with a small driving force (10.1021/jacs.9b01465). In fact, what the authors describe as multiply charge transfer states may be a superposition of NFA exciton and CT state absorption and emission. That the different peaks in the EL spectra have different origins is also suggested by the bias dependent EL spectra in Figure S16. Therefore, a proper decomposition of the spectra in the contributions from CT and singlet states is mandatory. This is usually done by recording the EQE and EL of the neat NFA layers. With such a proper deconvolution, an exact determination of the CT energy will be feasible.

Answer: We thank the reviewer for the critical comment. We agree that the electroluminescence emission are superposition of singlet and charge transfer states. We fabricated devices with only acceptor film as the active layer (ITO/ZnO/acceptor/MoOx/Ag) and measured the EL of neat NFA layers. Special attention needs to be paid during processing in order to suppress strong crystallization of the acceptors which result in rough films and failure of the devices. After optimizing the film formation conditions of solution and substrate temperature (hot chloroform solution and substrate temperature at 60 °C) as well as spin speed (both 1000 and 3000 rpm), we managed to obtain EL of devices based on IDTT-C8-TIC and IDTT-C10-TIC, while the devices based on IDTT-C6-TIC have sever leakage current and failed for EL measurement.

The results are shown below (Figure R1). Based on the emission of the acceptors, we can assign the emission peaks in the devices based on blend films. For IDTT-C8-TIC and IDTT-C10-TIC based solar cells, the narrower peaks at higher energy could be attributed to emission from singlet states with vibrational features, while the extra peaks in the blend at lower energy are attributed to emission from interfacial charge transfer states.

Figure R1. Normalized EL and corresponding fits for devices based on pristine acceptor films.

In order to determine the charge transfer state energy, we tried to decompose the EL spectra by emission from singlet states of acceptor excitons and from charge transfer states as the reviewer mentioned. It is noted that Si CCD detector is used in PL and EL measurement. A general way to correct spectra is to divide the measured signal by detector sensitivity, which has been performed for all the EL and PL spectra reported in the original manuscript. While the detector has good sensitivity in the range of acceptor exciton emission, the

sensitivity at lower energy part of charge transfer state emission below 1.3 eV (see the Figure R2) is low. After correction of the measured signal by dividing detector sensitivity, this lower energy part of charge transfer states gets much more pronounced. Although this detector sensitivity correction is not suitable for the low energy part, it is necessary for the whole spectra in order to get correct peak positions. So we used the corrected spectra for decomposition of emissions from acceptor excitons and charge transfer states with the emission below 1.3 eV as one fitting peak to account for vibrations of the main charge transfer states emission peak (above 1.3 eV).

Figure R2. Comparison of electroluminescence spectra before and after correction with detector sensitivity.

Figure R3. EL peak decomposition. The EL spectra of blend were fitted using 4 Gaussian peaks according to the equation $y = y_0 + \sum_{i=1}^4 A_i * e^{-\frac{(x-x_{ci})^2}{2w_i^2}}$, y_0 is spectra offset, A is peak amplitude, x_c is the peak center energy, w is the width of the peak.

The blend spectra decomposition to emission from singlet acceptor exciton and charge transfer states is shown in the following Figure R3. The two high energy fitting peaks are assigned to singlet emission with vibrational features, and the two fitting peaks at lower

energy are from charge transfer states emission. The peak center energy and peak width of the fitting components (shown in the Table R1) are fitted with information from the not corrected spectra where the main CT state emission peak is dominant.

According to the Marcus theory, the spectra line shape of charge transfer state emission, $N(E)$, and absorption, $A(E)$, can be described by equation (1) and (2) (K. Vandewal, *Annu. Rev. Phys. Chem.* 2016. 67:113–33):

$$N(E) \propto E \exp\left[-\frac{(E_{CT}-\lambda-E)}{4\lambda k_B T}\right] \quad (1)$$

$$A(E) \propto \frac{1}{E} \exp\left[-\frac{(E_{CT}+\lambda-E)}{4\lambda k_B T}\right] \quad (2)$$

where E is energy, E_{CT} is charge transfer state energy and λ is reorganization energy. To determine the charge transfer state energy, we further validated the E_{CT} and λ from EL spectra by simultaneously fitting EL and FTPS spectra according to the Marcus theory (Figure R4). It is shown that the E_{CT} and λ obtained from fitting of the main peak of charge transfer state emission can nicely fit the pronounced absorption shoulder of FTPS spectra. Charge transfer states energy for all the three blends and their comparison with V_{OC} are shown in the Table R2. It is shown that the three systems have different charge transfer states energy of 1.37 eV, 1.41 eV, and 1.50 eV for IDTT-C6-TIC, IDTT-C8-TIC and IDTT-C10-TIC based solar cells respectively. The differences in E_{CT} largely explained V_{OC} variations.

Figure R5. Fitting of charge transfer state emission ($N(E)$) and absorption ($A(E)$) according to Marcus theory for solar cells based on (a) IDTT-C6-TIC, (b) IDTT-C8-TIC and (c) IDTT-C10-TIC. Red line is reduced emission spectra, orange dots line are the reduced absorption spectra, dark and light gray line are the corresponding fits of the high energy part of charge transfer state emission and lower energy part of charge transfer state absorption. Charge transfer state energy (E_{CT}) and reorganization energy (λ) are indicated in the graph inset.

Table R1. Peak center energy and peak width of the fitting components (Gaussian peaks).

Acceptor	IDTT-C6-TIC	IDTT-C8-TIC	IDTT-C10-TIC
x_{c1} (eV)	1.20 ± 0.001	1.21 ± 0.001	1.31 ± 0.004
w_1 (eV)	0.065 ± 0.000	0.08 ± 0.001	0.075 ± 0.004
x_{c2} (eV)	1.33 ± 0.000	1.33 ± 0.001	1.45 ± 0.001
w_2 (eV)	0.050 ± 0.000	0.065 ± 0.001	0.05 ± 0.002
x_{c3} (eV)	1.47 ± 0.002	1.43 ± 0.002	1.54 ± 0.002
w_3 (eV)	0.040 ± 0.004	0.63 ± 0.001	0.03 ± 0.002
x_{c4} (eV)	1.55 ± 0.048	1.57 ± 0.001	1.62 ± 0.000
w_4 (eV)	0.072 ± 0.026	0.047 ± 0.001	0.03 ± 0.000

Table R2. Charge transfer state energy and corresponding V_{OC} and $V_{OC, nr}$ of the corresponding solar cells.

Acceptor	E_{CT} (eV)	LUMO	V_{OC} (V)	$V_{OC, nr}$ (V)
IDTT-C6-TIC	1.37	-3.99	0.85	0.33
IDTT-C8-TIC	1.41	-3.97	0.88	0.32
IDTT-C10-TIC	1.50	-3.91	0.98	0.26

3. Finally, a link between the different crystal structures and the CT (V_{oc}) properties is missing. Is the increase of the CT energy with increasing side chain length simply related to an increase in the LUMO energy of the NDA (which would be a trivial and well-known fact)? Or are there additional factors (relative orientation etc.) contributing to the interplay between acceptor molecular packing and CT energy. Here, I remind the authors that the CT energy and decay kinetics is primary determined by the properties of the DA interface, and that there are numerous mechanisms how these properties are affected by the packing and orientation of the molecules in the individual phases.

Answer: We thank the reviewer for the critical point. We identified two major differences in CT state for the investigated systems. First, the energy differences in E_{CT} are closely related to the LUMO energy differences. We have compared the LUMO of the acceptors (see Table R2) and the E_{CT} derived from simultaneously fitting EL and FTPS spectra. The energy difference between donor HOMO with acceptor LUMO showed an order of IDTT-C6-TIC < IDTT-C8-TIC < IDTT-C10-TIC, which corresponds to the same trend for charge transfer state energy. This result indicates that the differences in CT state energy can be at least partially due to LUMO energy differences among the different acceptors. Secondly, the spectra line shape is significantly different (see Figure R2-3), which is most probably originated from molecular packing.

4. The captions of the Figures in the SI are short and often do not provide sufficient information. E.g. it is not stated whether the crystal structures in Figures 5-11 are from X-ray diffraction experiments on single crystals, from MD simulations, or from a combination of both.

Answer: All structure pictures are drawn from single crystal information files, which are measured by single crystal X-ray diffraction. MD simulation has also been done based on the single crystal structures. The cation and crystallographic axes have been added into supporting information.

5. Figure S1 and Table S1: In my opinion, the values of the LUMO in Table S1 do not correspond to the CV curves in Figure S1. For example, IDTT-C8-TIC displays the most negative onset of the redox process, which is in contrast to the values in Table S1. I was also surprised of the fact that the reduction is not reversible for any of the three NFAs (which makes an exact determination of the LUMO energy nearly impossible). A detailed discussion of the JCV curves and their analysis is needed.

Answer: The CV curves of IDTT-CX-TIC in Figure R5 have been now vertically translated and the corresponding tangents have also been drawn for better indication. The energy levels are calculated according to the following equations:

$$E_{\text{HOMO}} = -[(4.8 - E_{\text{Fc}/\text{Fc}^+}) + E_{\text{ox}}] \text{ (eV)}$$

$$E_{\text{LUMO}} = -[(4.8 - E_{\text{Fc}/\text{Fc}^+}) + E_{\text{red}}] \text{ (eV)}$$

$$E_{\text{Fc}/\text{Fc}^+} = 0.43 \text{ eV}$$

We take E_{HOMO} of IDTT-C8-TIC as an example:

$$E_{\text{HOMO}} = -[(4.8 - 0.43) + 1.27] = -5.64 \text{ (eV)}$$

Figure R5. Cyclic voltammetry curves of **a** IDTT-CX-TIC and **b** Ferrocene. **c** Normalized thin-film absorption of IDTT-CX-TIC.

Then we can obtain a series of values as below. All of them are the same as those reported in Supplementary Table S1.

We also noticed that the CV curves exhibit irreversible reduction reaction for IDTT-CX-TIC and other 1,1-dicyanomethylene-3-indanone based NFAs. We think that such property is probably due to the strong electron-withdrawing ability of 1,1-dicyanomethylene-3-indanone. The similar phenomena have been reported in the literature (see Adv. Mater. 2015, 27, 1170; Adv. Mater. 2017, 1705208; Adv. Mater. 2017, 1702125; Adv. Mater. 2017, 1703080; Adv. Mater. 2018, 1707150; Sci. Bull., 2017, 62, 1494; Joule, 2019, 3, 1140).

6. The directions of the crystallographic axes should be added to Figure S5-11 for a better understanding of the molecular packing.

Answer: The cation and crystallographic axes have been added into the supporting information.

7. Figure S46 and Table S2. According to the methods section, the current density–voltage (J – V) characteristics of the hole or electron only devices were fitted by the Mott–Gurney law. According to this law, the current increases strictly quadratically with voltage, meaning that the slope of the JV curves (in the log-log plots in Figure S36) should all have the same slope of 2. This is clearly not the case. A detailed discussion of the JV curves and their analysis is needed.

Answer: We agree with the reviewer that according to the Mott–Gurney law, the slope of the J – V curves (in the log-log plots) should be equal to 2. In order to reconsider both hole and electron mobility, we refabricated SCLC devices to attain ohmic region and SCLC region as a more reliable charge transport study, as shown in Figure R6. And the calculated mobilities are summarized in Table R3.

Figure R6. Characteristic curves of (a) hole-only SCLC devices, and (b) electron-only SCLC devices.

Table R3. Mobilities of neat IDTT-CX-TIC and PBT1-C:IDTT-CX-TIC films.

μ_e ($\text{cm}^2 \text{V}^{-1} \text{s}^{-1}$)	Electron mobility	Hole mobility
IDTT-C6-TIC	1.2×10^{-3}	
IDTT-C8-TIC	2.2×10^{-4}	
IDTT-C10-TIC	9.8×10^{-5}	
PBT1-C:IDTT-C6-TIC	2.0×10^{-4}	7.3×10^{-4}
PBT1-C:IDTT-C8-TIC	1.2×10^{-4}	6.5×10^{-4}
PBT1-C:IDTT-C10-TIC	6.7×10^{-5}	3.0×10^{-4}

8. in my opinion, the TAS data are superfluous as they do not contribute to the understanding of the interplay between molecular packing and the CT properties. Moreover, the results from TAS (where all three systems display similar exciton lifetimes and similar charge formation kinetics) are in contradiction to the structural data (Figure 3 and corresponding text), where is stated that the large scale phase separation of the PBT1-C:IDTT-C6-TIC blend “is detrimental to efficient charge transfer”.

Answer: Thanks for the valuable feedback from the reviewer, which prompted us to clarify the contribution of TAS in the manuscript as follows. The main purpose of this manuscript is to demonstrate how molecular packing of non-fullerene acceptors can influence charge transport and photovoltaic performance. Previous efforts to tune charge transport in organic semiconductors have often come at the expense of exciton transport, prompting us to investigate via TAS whether these materials suffered this compromise. Notably, we found that exciton diffusion is always efficient and apparently insensitive to the solid-state packing of IDTT-CX-TIC, which we rationalise on the basis that excitons have sufficiently strong resonant overlap to hop a long distance, rendering the local packing structure less relevant.

We have substantially revised the TAS section along these lines, including adding new references to describe the previous understanding of local packing effects on exciton transport.

The reviewer is also correct that the previous statement about 200 nm domain size is at odds with the exciton transport measurements. This discrepancy reflects the nuance around the phase size dispersion, which we have clarified as follows: “A broad scattering was detected in RSoXS, corresponding to a wide distribution of phase separation lengths with the largest domain size of about 200 nm. These large phases will suffer inefficient charge transfer, leading to suppressed short-circuit current (J_{sc}) and fill factor (FF) in OSCs.”

In the TAS section, we also expanded the point as follows: “We find that shorter exciton lifetimes in the blend compared with the neat NFA films (Supplementary Fig. 18), are consistent with efficient exciton quenching in all cases, and considering the high exciton diffusion coefficients of NFAs, this could still correspond to phase sizes of 10s of nanometers. We expect that there may be residual unquenched excitons in the IDTT-C6-TIC blend containing some large phases, however that signature would be overwhelmed by the majority of excitons undergoing distinctly faster charge generation.”

Finally, the control over molecular packing provides an excellent system for a more in-depth investigation of photophysics, particularly focusing on configurational effects of charge separation. We plan to follow up with such studies, which are beyond the scope of the present manuscript.

Reviewer #3 (Remarks to the Author):

In this work, the authors rationally designed non-fullerene acceptor (NFA) molecules by changing the side chain length to realize different molecular stacking modes and their impact on charge transfer dynamics, non-radiative recombination loss and overall device performance.

The authors systematically studied molecular structures of NFAs and modelled their potential stacking modes. They showed that end group-facilitated stacking and long-range order can be replaced with an intermixed stacking or even a non-stacking mode, with latter being overlooked by the community, to achieve superior performance.

Although there is some new understanding obtained through this work regarding to molecular design endeavor in the future, universality of this approach is yet to be demonstrated for other and more efficient (Y-series) NFAs, and novelty of investigating the influence of molecular engineering on photovoltaic performance is somewhat arguable here. I also believe some of the findings, such as the reason for the difference in photovoltaic performance (charge generation, charge transport etc.) are not backed conclusively. Overall, side chain engineering is not anymore considered as a novel strategy and I do not find the manuscript suitable for Nature Comm. caliber unfortunately.

Answer: We greatly thank the reviewer for her/his effort in evaluating our manuscript and for providing valuable comments. However, we politely disagree with the conclusion made by the reviewer that this work is not suitable for publication in Nature Communications. We'd like to take this opportunity to better explain the novelty and significance of our work.

We agree that side chain engineering is not anymore considered as a novel strategy, which has been widely used for the synthesis of different types of photovoltaic materials. However, in this work, we mainly investigate the effects of solid-state packing on the electronic structure and function of NFA molecules, and aim to establish a rational structure-property relationship to guide the development of next-generation NFA materials. The side chain engineering is not the focus of our work, but is merely used to manipulate the solid-state packing of NFAs. We interestingly find that the NFA backbone can be manipulated from a strong π - π stacking mode to an intermixed packing mode, and to a non-stacking mode by rationally modifying the length of the side chains. Beyond our current understanding, this work highlights that close atom contacts in a non-stacking mode can enable efficient charge carrier hopping transport through close side atom interactions. The optimized OSCs free from end group π - π stacking yield a superior PCE of 12.7% with reduce non-radiative recombination loss to that of OSCs rely on classic end group π - π stacking formed major transport channels, which has not been recognized and reported in the organic photovoltaic community. More importantly, molecular and crystal engineering allows the combination of the two solid-state packing motifs together in a BHJ blend, leading to a PCE of 13.7%, surpassing the single mode interaction dictated function. This work stirs a new concept in

advanced molecules design for next-generation organic photovoltaics, which is of great importance and interests to the research community.

In general, the article is well written with an elaborate language and few minor mistakes (typos, wrong wording, repetition of words with similar meanings etc.) to be corrected here and there (also in Supplementary Information). I would also suggest the use of shorter sentences in some parts for a better read.

Answer: We corrected the writing mistakes and shortened some sentences for a better read in the revised manuscript.

Comment 1: Power conversion efficiencies (PCEs) reported for NFA organic solar cells are now above 17% (>18% in Science Bulletin). It would be better, if authors update the information on Page 4.

Answer: We updated the reference and cited some representative works in which the PCEs of over 17% are reported.

Comment 2: It is not clear to me whether molecular stacking manipulation in NFAs is challenging. Initially, it is indicated that obtaining single crystals is difficult. Then, in the very next sentence, it is said chemistry can solve such problems which is regarded as molecular engineering and widely used by scientists to synthesize analogues of archetypal NFAs such as Y6 for various improvements in organic solar cell (OSC) performance.

Answer: The difficulties should be separated into two categories. First, the synthesis of new materials is a chemistry effort, which for sure takes much effort. Second, the growth of single crystal is a tedious and time consuming practice, which cannot always be successful. These two factors add up together lead to the difficulty in obtaining the single crystal of NFA materials. We started the project by trying to understand the molecular packing of the NFA we worked on. To ensure the solubility of NFAs in the common solvents, bulky side-chains are typically used to attach to the donor core of NFAs. Due to the steric hindrance of the bulky side chains, it is difficult to grow the single crystals of NFAs. For example, the single

crystal of star molecule ITIC was finally achieved at ambient temperature by the slow solvent vapor diffusion approach for 3–5 days, and it is polymorph. We systematically changed the backbone, and quite lucky to obtained single crystals. We think the success is due to the chemistry of using linear side chains, which is much easier to order and crystallize. And then we started to think the correlation between NFA function and crystalline packing. Regarding the Y6 molecule, due to its large backbone size and bulky side chains, only few reference reported its single crystal. The rarity of single crystal report indicate the difficulties in single crystal growth.

Comment 3: It would be a better guide to the eye, if the backbone, end groups and side chains of molecules were in different colors in Sup Fig 8 and 10 like presented in Sup Fig 3. It is hard to follow what is depicted in the figures in their current states. Also, incorporating the interaction types observed in IDTT-C8-TIC and IDTT-C10-TIC into respective figures in SI would make it easier to follow.

Answer: The structure pictures in supporting information have been updated.

Comment 4: Is there a particular rationale behind using IT core for the molecules reported in this study rather than electron-deficient core adopted in Y-series non-fullerene acceptors which outperform this class of NFAs? Is this alkyl chain-dependent packing mode engineering a universally applicable approach for other NFAs?

Answer: Due to its large backbone size and bulky side chains, it is quite challenging to attain the single crystal of Y6 molecule. Therefore, we here used the IT core with the linear side chains to increase molecular crystallinity and then we can easily obtain all the single crystals of IDTT-CX-TIC.

It is well established that side chain substituents influence the packing motif, electronic structure, and charge transport properties of the crystals of NFAs (ChemPhysChem 2019, 20, 2608). Thus we think that side-chain engineering is a universally applicable approach for other NFAs to tune their packing mode.

Comment 5: Crystal packing and morphology investigation section could use few references regarding to assessment of π - π stacking distances, how alkyl chains impact charge transfer

process and their crystallization the molecular assembly, and estimation of crystal coherence lengths on Page 10 and 11.

Answer: According to the reviewer's suggestion, we cited several references that correlate with the crystal packing and morphology investigation,

Comment 6: The electron mobility of PBT1-C:IDTT-C10-TIC is on the order of 10^{-4} in Table 1, in contrast to what is stated on Page 13 (on the order of 10^{-5}). Also, based on Table 1 and Supplementary Table 2, electron mobility of the blend is higher than that of neat acceptor film which is surprising.

Answer: We rechecked the mobility data. As shown in Table 1 (see the originally submitted manuscript), the electron mobility of PBT1-C:IDTT-C10-TIC is $(0.7 \pm 0.1) \times 10^{-4} \text{ cm}^2 \text{ V}^{-1} \text{ s}^{-1}$, which is not higher than that ($7.8 \times 10^{-5} \text{ cm}^2 \text{ V}^{-1} \text{ s}^{-1}$) of neat film. In order to reconsider both hole and electron mobility, we refabricated SCLC devices to attain ohmic region and SCLC region as a more reliable charge transport study, as shown in Figure R6 and Table R3. The calculated mobilities are of PBT1-C:IDTT-C10-TIC is $6.7 \times 10^{-5} \text{ cm}^2 \text{ V}^{-1} \text{ s}^{-1}$, which is smaller than that ($9.8 \times 10^{-5} \text{ cm}^2 \text{ V}^{-1} \text{ s}^{-1}$) of neat film.

Comment 7: Could authors expand on the influence of the molecular ordering of NFAs on V_{oc} s? How exactly does the microstructure of BHJ blends govern the V_{oc} ?

Answer: It is notable that the V_{oc} is not only governed by the materials properties, but also by the device configuration, processing conditions, and others. Nevertheless, as the solar cells were fabricated under the same conditions in the same device configuration, the solid state molecular packing could be the major factor that contributes to the V_{oc} change. In our current work, the packing motif without obvious pi-pi stacking (IDTT-C10-TIC) leads to a high V_{oc} , which is mainly due to its higher CT state energy and lower non-radiative recombination losses. Moreover, the CT state energy is strongly influenced by the D/A interface properties, which cannot be experimentally resolved so far (as detailed previously – 1st question of Reviewer 2). As such, we cannot give a clear conclusion on how exactly the microstructure of BHJ blends govern the V_{oc} of NFA solar cells.

Comment 8: I am wondering how Eloss is calculated in units of V, where the unit of E_g is eV in Table 2. One way to correct this would be multiplying each type of VOC with q to convert the units to eV.

Answer: We thank the reviewer's remind. The unit has been converted to eV.

Comment 9: Do authors have any comments on how PBT1-C:IDTT-C6-TIC blend system exhibits competitive, if not superior, electron and hole mobilities despite its large aggregates and large-scale phase separation compared to other two blend systems? If this is an expected outcome, why are the JSC and FF significantly lower? According to the findings of this study, exciton diffusion is not drastically affected by the crystal structures and molecular packings either. The work cited at the end of Results section mainly investigates the IDIC molecule, whereas this study looks into 3 different molecules purposefully engineered to observe the effect of side chain length on molecular packing and eventually, non-radiative VOC loss. Thus, it would be more conclusive if there was direct evidence of low energetic disorder and self-spectral overlap for respective systems presented here.

Answer: The strong aggregates and phase separation in tightly packed PBT1-C:IDTT-C6-TIC blends induces high electron and hole mobilities, which however do not guarantee good performance in organic solar cells. The device performance is a collective output from materials properties, thin film morphology and device engineering. However, IDTT-C6-TIC blends didn't perform well in the device configuration used in this work, compared to the other two NFA blends. We referenced IDIC work mainly due to its close relevance. There is no such report for a similar backbone NFA in the current stage. The main purpose, as we have elaborated, is to show the new effect of different molecular packing on the electronic and morphological changes, which correlated well with device performances.

Comment 10: I would suggest an edit for the sentence on Page 14 as "... the lowest CT state energy increases with increasing the side chain length, ...". In its current version of "... the lowest CT states energy in each system is increased with increasing the length of side

chain, ...”, it is read like each blend system has also a variation of side chain length in itself, in fact the side chain length is what separates each blend system.

Answer: We thank the reviewer’s comment. The sentence has been revised accordingly.

Comment 11: Although the non-radiative VOC losses were finetuned by side chain length engineering, the BHJ blend with the lowest non-radiative VOC loss is not the most efficient solar cell in this study. Hence, I am not sure how sound the message that is tried to get across to by the authors at the bottom of Page 14 and top of Page 15. Evidently, reducing the non-radiative VOC loss by designing molecules free from π - π stacking appears promising, if it also leads to higher efficiencies compared to all the other charge transfer modes.

Answer: Currently, relatively low V_{oc} has been one of the largest bottlenecks in determining the upper limit of PCEs of OSCs. We here demonstrate the importance of designing novel molecular semiconductors free from π - π stacking to further reduce the non-radiative recombination losses of OSCs. It should be noted that reducing the non-radiative recombination loss can lead to improved V_{oc} . However, the PCE of OSCs is determined by three parameters including V_{oc} , J_{sc} , and FF. The J_{sc} , and FF are strongly influenced by the light absorption, charge transport, charge recombination, etc. We believe that by further molecular engineering of NFAs and better morphology control, we can simultaneously realize low non-radiative voltage losses and high PCE.

REVIEWER COMMENTS

Reviewer #1 (Remarks to the Author):

The authors have addressed the comments from the reviewers and the paper can be accepted now.

Reviewer #2 (Remarks to the Author):

Sorry for the long delay in replying to the revised version. This was because of travelling and extensive administrative work due to Corona.

The authors provided a detailed reply to my questions and remarks, and made important changes to the document. The newly added data (and remeasured SCLC) and the more extensive analysis of the emission properties considerably improve the quality of the manuscript, which I now recommend for publication.

Regarding my first comment about the "interplay between molecular arrangement and orientation on the Voc and on the non-radiative losses." I agree with the authors there is currently no experimental tool to probe the microscopic structural properties of the DA interface. Still, a more detailed analysis of the GIWAXS data of the blend in comparison to the neat layers would be useful. For example, is the orientational distribution of the donor and acceptor domains different in the blend and the neat films? And how do coherence lengths in the neat film and blend compare.

Also, there is still a minor discrepancy between the statement on page 12 that for PBT1-C:IDTT-C6-TIC "These large phases will suffer inefficient charge transfer, leading to suppressed short-circuit current (JSC) and fill factor (FF) in OSCs " and the interpretation from TAS that exciton quenching is efficient in all cases. If the later statement holds, why is charge transfer inefficient and limits the Jsc?

All other comments in my report have been properly taken care of.

Reviewer #3 (Remarks to the Author):

The authors responded to most of the comments, yet I still do find the novelty argument not-convincing. I made point-by-point responses to the authors-find below. In addition, I do see some part is deleted or removed etc. about the last comment which I cannot find anymore in the discussions. It would be great if authors can explain it too.

(Remarks to the Author):

In this work, the authors rationally designed non-fullerene acceptor (NFA) molecules by changing the side chain length to realize different molecular stacking modes and their impact on charge transfer dynamics, non-radiative recombination loss and overall device performance.

The authors systematically studied molecular structures of NFAs and modelled their potential stacking modes. They showed that end group-facilitated stacking and long-range order can be replaced with an intermixed stacking or even a non-stacking mode, with latter being overlooked by the community, to achieve superior performance.

Although there is some new understanding obtained through this work regarding to molecular

design endeavor in the future, universality of this approach is yet to be demonstrated for other and more efficient (Y-series) NFAs, and novelty of investigating the influence of molecular engineering on photovoltaic performance is somewhat arguable here. I also believe some of the findings, such as the reason for the difference in photovoltaic performance (charge generation, charge transport etc.) are not backed conclusively. Overall, side chain engineering is not anymore considered as a novel strategy and I do not find the manuscript suitable for Nature Comm. caliber unfortunately.

Answer: We greatly thank the reviewer for her/his effort in evaluating our manuscript and for providing valuable comments. However, we politely disagree with the conclusion made by the reviewer that this work is not suitable for publication in Nature Communications. We'd like to take this opportunity to better explain the novelty and significance of our work.

We agree that side chain engineering is not anymore considered as a novel strategy, which has been widely used for the synthesis of different types of photovoltaic materials. However, in this work, we mainly investigate the effects of solid-state packing on the electronic structure and function of NFA molecules, and aim to establish a rational structure-property

relationship to guide the development of next-generation NFA materials. The side chain engineering is not the focus of our work, but is merely used to manipulate the solid-state packing of NFAs. We interestingly find that the NFA backbone can be manipulated from a strong π - π stacking mode to an intermixed packing mode, and to a non-stacking mode by rationally modifying the length of the side chains. Beyond our current understanding, this work highlights that close atom contacts in a non-stacking mode can enable efficient charge carrier hopping transport through close side atom interactions. The optimized OSCs free from end group π - π stacking yield a superior PCE of 12.7% with reduce non-radiative recombination loss to that of OSCs rely on classic end group π - π stacking formed major transport channels, which has not been recognized and reported in the organic photovoltaic community. More importantly, molecular and crystal engineering allows the combination of the two solid-state packing motifs together in a BHJ blend, leading to a PCE of 13.7%, surpassing the single mode interaction dictated function. This work stirs a new concept in advanced molecules design for next-generation organic photovoltaics, which is of great importance and interests to the research community.

Reviewer response: As stated in the initial comment, the new understanding obtained from this study is indeed interesting and may open new avenues for high-efficiency OPV material design, yet the universality of this approach is missing in the current state of the manuscript which makes it fall short of Nat Comm caliber impact. I strongly believe the finding presented in this study should be observed in other NFA systems with different cores for publication in a prestigious journal such as Nat Comm.

In general, the article is well written with an elaborate language and few minor mistakes (typos, wrong wording, repetition of words with similar meanings etc.) to be corrected here and there (also in Supplementary Information). I would also suggest the use of shorter sentences in some parts for a better read.

Answer: We corrected the writing mistakes and shortened some sentences for a better read in the revised manuscript.

Comment 1: Power conversion efficiencies (PCEs) reported for NFA organic solar cells are now above 17% (>18% in Science Bulletin). It would be better, if authors update the information on Page 4.

Answer: We updated the reference and cited some representative works in which the PCEs of over 17% are reported.

Reviewer response: The main text still reads as "over 16%" despite the updated references. I believe stating the highest reported efficiency in the manuscript would make it more up to date.

Comment 2: It is not clear to me whether molecular stacking manipulation in NFAs is challenging. Initially, it is indicated that obtaining single crystals is difficult. Then, in the very next sentence, it is said chemistry can solve such problems which is regarded as molecular engineering and widely used by scientists to synthesize analogues of archetypal NFAs such as Y6 for various improvements in organic solar cell (OSC) performance.

Answer: The difficulties should be separated into two categories. First, the synthesis of new materials is a chemistry effort, which for sure takes much effort. Second, the growth of single crystal is a tedious and time consuming practice, which cannot always be successful. These two factors add up together lead to the difficulty in obtaining the single crystal of NFA materials. We started the project by trying to understand the molecular packing of the NFA we worked on. To ensure the solubility of NFAs in the common solvents, bulky side-chains are typically used to attach to the donor core of NFAs. Due to the steric hindrance of the bulky side chains, it is difficult to grow the single crystals of NFAs. For example, the single crystal of star molecule ITIC was finally achieved at ambient temperature by the slow solvent vapor diffusion approach for 3–5 days, and it is polymorph. We systematically changed the backbone, and quite lucky to obtain single crystals. We think the success is due to the chemistry of using linear side chains, which is much easier to order and crystallize. And then we started to think the correlation between NFA function and crystalline packing. Regarding the Y6 molecule, due to its large backbone size and bulky side chains, only few references reported its single crystal. The rarity of single crystal report indicates the difficulties in single crystal growth.

Reviewer response: I am a bit confused about this response. The authors initially emphasized that the molecules were rationally designed to study the impact of the molecular packing on photovoltaic performance, but “And then we started to think the correlation between NFA function and crystalline packing.” sounds like they kept changing the backbone until they were able to realize single crystals, which rather feels like trial and error than a rational design strategy.

Comment 3: It would be a better guide to the eye, if the backbone, end groups and side chains of molecules were in different colors in Sup Fig 8 and 10 like presented in Sup Fig 3. It is hard to follow what is depicted in the figures in their current states. Also, incorporating the interaction types observed in IDTT-C8-TIC and IDTT-C10-TIC into respective figures in SI would make it easier to follow.

Answer: The structure pictures in supporting information have been updated.

Comment 4: Is there a particular rationale behind using IT core for the molecules reported in this study rather than electron-deficient core adopted in Y-series non-fullerene acceptors which outperform this class of NFAs? Is this alkyl chain-dependent packing mode engineering a universally applicable approach for other NFAs?

Answer: Due to its large backbone size and bulky side chains, it is quite challenging to attain the single crystal of Y6 molecule. Therefore, we here used the IT core with the linear side chains to increase molecular crystallinity and then we can easily obtain all the single crystals of IDTT-CX-TIC.

It is well established that side chain substituents influence the packing motif, electronic structure, and charge transport properties of the crystals of NFAs (ChemPhysChem 2019, 20, 2608). Thus we think that side-chain engineering is a universally applicable approach for other NFAs to tune their packing mode.

Reviewer response: This is the very reason I asked this question. It is not trivial or maybe not even possible to exploit this approach in other NFA molecules, as the authors also stated in the case of Y6 molecule. Additionally, the article the authors mention in their response is again mainly based on IDTT acceptors and it actually suggests the importance of formation of π -interaction for efficient charge transport which contradicts the finding communicated in this manuscript, although a wide variety of IDTT acceptors are studied. Thus, it still remains unclear if this approach can be utilized in NFAs with different cores.

Comment 5: Crystal packing and morphology investigation section could use few references regarding to assessment of π - π stacking distances, how alkyl chains impact charge transfer process and their crystallization the molecular assembly, and estimation of crystal coherence lengths on Page 10 and 11.

Answer: According to the reviewer's suggestion, we cited several references that correlate with the crystal packing and morphology investigation,

Comment 6: The electron mobility of PBT1-C:IDTT-C10-TIC is on the order of 10^{-4} in Table 1, in contrast to what is stated on Page 13 (on the order of 10^{-5}). Also, based on Table 1 and Supplementary Table 2, electron mobility of the blend is higher than that of neat

acceptor film which is surprising.

Answer: We rechecked the mobility data. As shown in Table 1 (see the originally submitted manuscript), the electron mobility of PBT1-C:IDTT-C10-TIC is $(0.7 \pm 0.1) \times 10^{-4} \text{ cm}^2 \text{ V}^{-1} \text{ s}^{-1}$, which is not higher than that ($7.8 \times 10^{-5} \text{ cm}^2 \text{ V}^{-1} \text{ s}^{-1}$) of neat film. In order to reconsider both hole and electron mobility, we refabricated SCLC devices to attain ohmic region and SCLC region as a more reliable charge transport study, as shown in Figure R6 and Table R3. The calculated mobilities are of PBT1-C:IDTT-C10-TIC is $6.7 \times 10^{-5} \text{ cm}^2 \text{ V}^{-1} \text{ s}^{-1}$, which is smaller than that ($9.8 \times 10^{-5} \text{ cm}^2 \text{ V}^{-1} \text{ s}^{-1}$) of neat film.

Reviewer response: The authors are correct about the mobility comparison. However, it would cause less of a confusion for readers, if they stated a value as $7 \times 10^{-5} \text{ cm}^2 \text{ V}^{-1} \text{ s}^{-1}$ rather than $0.7 \times 10^{-4} \text{ cm}^2 \text{ V}^{-1} \text{ s}^{-1}$ which defeats the purpose of using the exponential. We generally tend to look at the order of magnitude of mobility values rather than the factor. The updated mobility value for PBT1-C:IDTT-C10-TIC appears to be on the order of 10^{-4} in Table 1. I assume this is a typo.

Comment 7: Could authors expand on the influence of the molecular ordering of NFAs on VOCs? How exactly does the microstructure of BHJ blends govern the VOC?

Answer: It is notable that the Voc is not only governed by the materials properties, but also by the device configuration, processing conditions, and others. Nevertheless, as the solar cells were fabricated under the same conditions in the same device configuration, the solid state molecular packing could be the major factor that contributes to the Voc change. In our current work, the packing motif without obvious pi-pi stacking (IDTT-C10-TIC) leads to a high Voc, which is mainly due to its higher CT state energy and lower non-radiative recombination losses. Moreover, the CT state energy is strongly influenced by the D/A interface properties, which cannot be experimentally resolved so far (as detailed previously – 1st question of Reviewer 2). As such, we cannot give a clear conclusion on how exactly the microstructure of BHJ blends govern the Voc of NFA solar cells.

Reviewer response: The CT state energy is indeed influenced by, but not limited to, the D/A interface, as Reviewer 2 has already mentioned. My concern about the statement “We found that the open-circuit voltage (VOC) of OSCs is influenced by the microstructure of BHJ blends, i.e. the molecular ordering of NFAs.” in the manuscript is about basing the Voc on the microstructure of BHJ blends, although it is not possible to probe the D/A interface with our current understanding, as you responded to Reviewer 2. Therefore, I wonder how reliably one can correlate these two to each other with knowing that they actually cannot experimentally show it at least for the time being.

Comment 8: I am wondering how Eloss is calculated in units of V, where the unit of Eg is eV in Table 2. One way to correct this would be multiplying each type of VOC with q to convert the units to eV.

Answer: We thank the reviewer’s remind. The unit has been converted to eV.

Comment 9: Do authors have any comments on how PBT1-C:IDTT-C6-TIC blend system exhibits competitive, if not superior, electron and hole mobilities despite its large aggregates and large-scale phase separation compared to other two blend systems? If this is an expected outcome, why are the JSC and FF significantly lower? According to the findings of this study, exciton diffusion is not drastically affected by the crystal structures and molecular packings either. The work cited at the end of Results section mainly investigates the IDIC molecule, whereas this study looks into 3 different molecules purposefully engineered to observe the effect of side chain length on molecular packing and eventually, non-radiative VOC loss. Thus, it would be more conclusive if there was direct evidence of low energetic disorder and self-spectral overlap for respective systems presented here.

Answer: The strong aggregates and phase separation in tightly packed PBT1-C:IDTT-C6-TIC blends induces high electron and hole mobilities, which however do not guarantee good performance in organic solar cells. The device performance is a collective output from materials properties, thin film morphology and device engineering. However, IDTT-C6-TIC blends didn’t perform well in the device configuration used in this work, compared to the other two NFA blends. We referenced IDIC work mainly due to its close relevance. There is no such report for a similar backbone NFA in the current stage. The main purpose, as we

have elaborated, is to show the new effect of different molecular packing on the electronic and morphological changes, which correlated well with device performances.

Reviewer response: I get that the purpose of this study is to show a new effect, but there are claims made based on other material systems and their blends. I would prefer to see a similar investigation done for the material systems in this study, especially given that the authors have observed a new and unrecognized trend in this work.

Comment 10: I would suggest an edit for the sentence on Page 14 as "... the lowest CT state energy increases with increasing the side chain length, ...". In its current version of "... the lowest CT states energy in each system is increased with increasing the length of side chain, ...", it is read like each blend system has also a variation of side chain length in itself, in fact the side chain length is what separates each blend system.

Answer: We thank the reviewer's comment. The sentence has been revised accordingly.

Comment 11: Although the non-radiative VOC losses were finetuned by side chain length engineering, the BHJ blend with the lowest non-radiative VOC loss is not the most efficient solar cell in this study. Hence, I am not sure how sound the message that is tried to get across to by the authors at the bottom of Page 14 and top of Page 15. Evidently, reducing the nonradiative

VOC loss by designing molecules free from π - π stacking appears promising, if it also leads to higher efficiencies compared to all the other charge transfer modes.

Answer: Currently, relatively low Voc has been one of the largest bottlenecks in determining the upper limit of PCEs of OSCs. We here demonstrate the importance of designing novel molecular semiconductors free from π - π stacking to further reduce the non-radiative recombination losses of OSCs. It should be noted that reducing the non-radiative recombination loss can lead to improved Voc. However, the PCE of OSCs is determined by three parameters including Voc, Jsc, and FF. The Jsc, and FF are strongly influenced by the light absorption, charge transport, charge recombination, etc. We believe that by further molecular engineering of NFAs and better morphology control, we can simultaneously realize low non-radiative voltage losses and high PCE.

Reviewer #1 (Remarks to the Author):

The authors have addressed the comments from the reviewers and the paper can be accepted now.

Answer: We greatly thank the reviewer for the very positive evaluation of our work.

Reviewer #2 (Remarks to the Author):

Sorry for the long delay in replying to the revised version. This was because of travelling and extensive administrative work due to Corona.

The authors provided a detailed reply to my questions and remarks, and made important changes to the document. The newly added data (and remeasured SCLC) and the more extensive analysis of the emission properties considerably improve the quality of the manuscript, which I now recommend for publication.

Answer: We greatly thank the reviewer for the very positive evaluation of our work.

Regarding my first comment about the “interplay between molecular arrangement and orientation on the Voc and on the non-radiative losses.” I agree with the authors there is currently no experimental tool to probe the microscopic structural properties of the DA interface. Still, a more detailed analysis of the GIWAXS data of the blend in comparison to the neat layers would be useful. For example, is the orientational distribution of the donor and acceptor domains different in the blend and the neat films? And how do coherence lengths in the neat film and blend compare.

Answer: Thanks for the comments. We did a more detailed comparison of NFA neat film and blend film, and provided these results in the revised manuscript, which is also provided below (see Table R1-3).

The polymer in blend film presented a thin ring around 0.28 \AA^{-1} , indicating a random orientation, and its crystal coherence length (CCL) is around 9 nm. The (001), (100), and (1-10) diffraction rings are seen in IDTT-C6-TIC neat film, indicating a random orientation. In the blend film, the (001) diffraction peak became narrower in the in-plane direction, and the (100) diffraction peak disappeared. The π - π stacking peak showed a quite broad distribution. Thus a random orientation is still taken in the blend film. The narrowed (001) diffraction indicated that the IDTT-C6-TIC

molecules could be tilted through the short axis of the backbone, which is quite different from that in neat film. The CCLs of IDTT-C6-TIC decreased from 14.3 nm for the neat film to 10.4 nm for the blend film.

In IDTT-C8-TIC neat film, the (01-1), (021), (121), (242) crystal planes indicate its face-on orientation. Regarding the blend film, only (01-1) plane can be obviously observed in the in-plane direction. The CCLs of IDTT-C8-TIC neat and blend film are 12.8 nm and 9 nm, respectively.

The IDTT-C10-TIC molecules show a face-on orientation in neat and blend films. We choose the out-of-plane (112) crystal plane to compare the CCL. The CCLs of IDTT-C10-TIC neat film is around 6.8 nm, which is larger than that (4.6 nm) in the blend film.

Table R1. GIWAXS data of IDTT-C6-TIC neat and blend films.

Neat film	Location (\AA^{-1}) and crystal plane	Area	CCL (nm)	Blend film	Location (\AA^{-1}) and crystal plane	Area	CCL (nm)
	0.396 (001)	11.67	14.272		0.281	40.84	9.547
	0.600 (100/010)	3.711	16.771	In plane	0.371 (001)	18.711	10.351
In plane	0.725 (1-10)	3.357	10.95		0.725 (1-10)	8.897	3.980
	1.644	3.594	5.981		1.893	10.553	3.229
	1.855	3.978	4.095				
	0.388 (001)	9.810	10.953			0.299	32.28
Out of plane	0.597 (100/010)	6.267	15.400	Out of plane	0.714 (1-10)	6.400	4.186
	0.730 (1-10)	8.019	10.624		1.744	67.818	1.293
	1.854	6.641	3.286		1.913	9.242	4.789

Table R2. GIWAXS data of IDTT-C8-TIC neat and blend films.

Neat film	Location (\AA^{-1}) and crystal plane	Area	CCL (nm)	Blend film	Location (\AA^{-1}) and crystal plane	Area	CCL (nm)
					0.281	29.078	9.694
In plane	0.380 (01-1)	5.001	12.758	In plane	0.374 (01-1)	18.632	8.985

					0.300	29.816	10.202
	0.623 (021)	3.364	14.381		0.354	2.492	8602
Out of plane	0.814 (121)	0.569	15.618	Out of plane	0.858	1.6801	11.198
	1.605 (242)	11.043	6.090		1.747	21.395	2.959
	1.889	73.904	1.771		1.924	27.812	3.767

Table R3. GIWAXS data of IDTT-C10-TIC neat and blend films.

Neat film	Location (\AA^{-1}) and crystal plane	Area	CCL (nm)	Blend film	Location (\AA^{-1}) and crystal plane	Area	CCL (nm)
In plane	0.799	3.365	11.060	In plane	0.281	40.414	10.428
	0.857	2.720	10.313		0.801	4.393	10.389
					0.306	91.367	7.349
Out of plane	0.854 (112)	23.268	6.785	Out of plane	0.854 (112)	46.411	4.595
	1.697 (224)	42.569	4.095		1.703	131.5	2.770

Also, there is still a minor discrepancy between the statement on page 12 that for PBT1-C:IDTT-C6-TIC “These large phases will suffer inefficient charge transfer, leading to suppressed short-circuit current (JSC) and fill factor (FF) in OSCs “ and the interpretation from TAS that exciton quenching is efficient in all cases. If the later statement holds, why is charge transfer inefficient and limits the Jsc?

Answer: We appreciate the referee making that point earlier, which prompted us to qualify the statement about TAS measurements of charge generation efficiency in the previous revision. The relevant statement that reconciles this apparent contradiction is on page 16:

'We expect that there may be residual unquenched excitons in the IDTT-C6-TIC blend containing some large phases, however that signature would be overwhelmed by the majority of excitons undergoing distinctly faster charge generation.'

All other comments in my report have been properly taken care of.

Reviewer #3 (Remarks to the Author):

The authors responded to most of the comments, yet I still do find the novelty argument not-convincing. I made point-by-point responses to the authors-find below. In addition, I do see some part is deleted or removed etc. about the last comment which I cannot find anymore in the discussions. It would be great if authors can explain it too.

Answer: We greatly thank the reviewer for this valuable comment. We apologize for any ambiguity caused, because the part of the manuscript related to the last comment was reorganized and rephrased with more quantitative discussions on energy of charge transfer state, as suggested by the 2nd reviewer. In short, based on more detailed analysis of the detector sensitivity limitations (Figure S15 in the revised manuscript) and added experimental results of the singlet emission of acceptors (Figure S16 in the revised manuscript), we have performed a multi Gaussian spectra deconvolution of EL spectra (Figure S17 in the revised manuscript) as well as simultaneous fitting of EL and FTPS spectra according to Marcus theory (Figure S18 in the revised manuscript), which enabled us to unambiguously determine the charge transfer state emission energy (added in revised Table 2). The charge transfer state energy can in a large extent explain the variations in V_{oc} according to energy gap law. We also found that the energy of charge transfer state is closely related to the LUMO energies of the acceptors. So we conclude that the different molecular packing due to side-chain engineering affected the energy level alignment at the D/A interface, and hence the CT state energy as well as V_{oc} losses.

Back to the reviewer's last comment, it is worthwhile to highlight again that, we demonstrate in this work the importance of designing novel molecular semiconductors free from π - π stacking to further reduce the non-radiative recombination losses. The relatively high non-radiative recombination loss is one of the largest bottlenecks in determining the upper limit of V_{oc} as well as PCE of OSCs. The designed material system free from π - π stacking was not able to achieve the highest efficiency owing to the slightly lower J_{sc} and FF, which are strongly influenced by the light absorption, charge transport, charge recombination processes, etc. Nevertheless, these processes are not determined by the concept reported in this work. We believe that by further molecular engineering of NFAs and better morphology control, we can simultaneously achieve both high V_{oc} and PCE.

The following part in the original manuscript is thus replaced/rephrased with more quantitative discussions based on our in-depth analysis of charge transfer state emission and absorption properties as shown in the revised supporting Figure S15-18.

'For all systems there are multiple charge-transfer (CT) states as seen in the EL spectra (Supplementary Figs. 16 and 17). It is worthwhile to highlight that the lowest CT states energy in each system is increased with increasing the length of side chain, which correlates well with the reduced non-radiative V_{oc} losses of OSCs. It was found that the non-radiative recombination losses in OSCs could be influenced by many factors, one of which is caused by electron-vibration coupling (intrinsic molecular vibration), which is fundamentally different from the mechanisms analyzed for inorganic semiconductors²⁹.'

As stated in the initial comment, the new understanding obtained from this study is indeed interesting and may open new avenues for high-efficiency OPV material design, yet the universality of this approach is missing in the current state of the manuscript which makes it fall short of Nat Comm caliber impact. I strongly believe the finding presented in this study should be observed in other NFA systems with different cores for publication in a prestigious journal such as Nat Comm.

Answer: We appreciated the valuable comments from the reviewer that the new understanding obtained from our study is indeed interesting and may open new avenues for high-efficiency OPV material design. The reviewer was wondering if the finding presented in our study could be observed in other NFA systems. This is in fact a good question that we are actually carrying on this approach in extending to new materials systems. We are glad to report to you that our preliminary results indicate that π - π stacking though providing an important aspect in NFA crystallization, it is not necessarily the only way. There are new systems that can be π - π stacking free, and still show high photovoltaic performance:

Figure R1. Single crystal structure of IDTT-C12-TIC. a) The backbone structure of IDTT-C12-TIC. b) The molecular conformer of IDTT-C12-TIC. The longer alkyl side chain is symmetrically folded along the backbone. c) and d) present the side chain separates IDTT-C12-TIC backbone layers, which leads to no π - π interactions.

IDTT-C12-TIC and IDTT-C10-TIC shares the same molecular skeleton, but exhibiting different side chain length. Similar to IDTT-C10-TIC, a large π - π stacking distance was recorded for IDTT-C12-TIC (see Figure R1). OSCs based on PBT1-C:IDTT-C10-TIC blend showed a PCE of 11.1%, with a V_{oc} of 0.97 V, a J_{sc} of 17.6 mA/cm², and a FF of 64.9%.

Figure R2. Single crystal structure of IDTT-C10-IC. a) The backbone structure of IDTT-C10-IC. It is the same with IT4F backbone. b) The molecular conformer of IDTT-C10-IC. The longer alkyl side chain is symmetrically folded along the backbone. c) The side chain separates IDTT-C10-IC backbone layers and forms no π - π structure. And two neighbor layers form a V type. d) Different IDTT-C10-IC layers (ring with different color) form hydrogen bonds and forms a lamellar structure.

IDTT-C10-IC and IDTT-C10-TIC share the same molecular skeleton and side-chain length, but with different ending group. As shown in Figure R2, the C10 alkyl side chain is symmetrically folded along backbone, which separates the backbone layers and leads to no π - π interactions. The as-cast device based on PM7:C10-IT4F blend showed a PCE of 11.5%, with a V_{oc} of 0.89 V, a J_{sc} of 18.3 mA/cm², and a FF of 70.4%. And the optimal device can yield a high PCE of 14.2%.

All the above results indicate that π - π stacking free could be a promising and universal approach enabling sufficient charge transfer efficiency in NFA systems. The related study is still ongoing. We believe that by further molecular engineering of NFAs and better morphology control, we can achieve high PCEs.

The main text still reads as “over 16%” despite the updated references. I believe stating the highest reported efficiency in the manuscript would make it more up to date.

Answer: We changed the efficiency value from 16% to 18% in the revised manuscript.

I am a bit confused about this response. The authors initially emphasized that the molecules were rationally designed to study the impact of the molecular packing on photovoltaic performance, but “And then we started to think the correlation between NFA function and crystalline packing.” sounds like they kept changing the backbone until they were able to realize single crystals, which rather feels like trial and error than a rational design strategy.

Answer: While we work on NFA systems, we continue to think about the function of NFA and its correlation of crystalline states. A global understanding would require large amplitude of materials and crystals of different chemical structures, which, however, is not a viable mission. We do start with commercially available materials to start crystal growth, since some of them have reported crystalline structures. It should also be noted that IDTT core is a structure that is

quite commonly used in NFA molecules, and it is also a chemical that is of easy access to us. And thus we start to convolute a research from the current backbone system we are familiar with. We systematically change the side chain of these materials, and we are lucky enough that these single crystals are obtained. We do see interesting results from the current materials systems that fulfill the needs of structure-property research. The rational design of materials, as appeared in the text, is more of calling the importance of understanding solid-state structure with respect to its chemistry. It is not an attempt to predict the best materials for function of crystallization study. The core focus in design is side-chain impact on molecular packing in solids, not a trial and error approach to find the best material that can crystallize.

This is the very reason I asked this question. It is not trivial or maybe not even possible to exploit this approach in other NFA molecules, as the authors also stated in the case of Y6 molecule. Additionally, the article the authors mention in their response is again mainly based on IDTT acceptors and it actually suggests the importance of formation of π -interaction for efficient charge transport which contradicts the finding communicated in this manuscript, although a wide variety of IDTT acceptors are studied. Thus, it still remains unclear if this approach can be utilized in NFAs with different cores.

Answer: Please see our aforementioned response regarding the universality of this approach. It can be seen clearly that this approach can be utilized in different NFAs.

The authors are correct about the mobility comparison. However, it would cause less of a confusion for readers, if they stated a value as $7 \times 10^{-5} \text{ cm}^2 \text{ V}^{-1} \text{ s}^{-1}$ rather than $0.7 \times 10^{-4} \text{ cm}^2 \text{ V}^{-1} \text{ s}^{-1}$ which defeats the purpose of using the exponential. We generally tend to look at the order of magnitude of mobility values rather than the factor. The updated mobility value for PBT1-C:IDTT-C10-TIC appears to be on the order of 10^{-4} in Table 1. I assume this is a typo.

Answer: The electron mobility of IDTT-C10-TIC in the blend is $6.7 \times 10^{-5} \text{ cm}^2 \text{ V}^{-1} \text{ s}^{-1}$. We corrected it in the revised manuscript.

The CT state energy is indeed influenced by, but not limited to, the D/A interface, as Reviewer 2 has already mentioned. My concern about the statement “We found that the open-circuit voltage (V_{oc}) of OSCs is influenced by the microstructure of BHJ blends, i.e. the molecular ordering of NFAs.” in the manuscript is about basing the V_{oc} on the microstructure of BHJ blends, although it is not possible to probe the D/A interface with our current understanding, as you responded to Reviewer 2. Therefore, I wonder how reliably one can correlate these two to each other with knowing that they actually cannot experimentally show it at least for the time being.

Answer: As it is raised, and we replied, V_{oc} , energy loss, morphology, chemical structure, and their detailed correlations are among the most difficult problems in the field of OPV due to the lack of characterization method. However, the real difficulty is the scientific logic behind and quantitative analysis. In a qualitative manner, we could still find they are highly connected. We see in this work that engineering of side-chains leads to electronic structure change in thin film, contributing partially from the mixed region and material properties. We could not obtain their weighted contribution for the time being, and thus the correlation is of little value quantitatively. However, the influence of solid-state packing of molecules, for those of the same backbone, and thus of the same intrinsic electronic structure molecularly, on the V_{oc} and energy loss can be clearly seen. And thus qualitatively, their connection can be established.

I get that the purpose of this study is to show a new effect, but there are claims made based on other material systems and their blends. I would prefer to see a similar investigation done for the material systems in this study, especially given that the authors have observed a new and unrecognized trend in this work.

Answer: The previous study cited in the manuscript showed that the high exciton diffusion of IDIC molecular acceptors is due to the good self-spectral overlap and low energetic disorder.

We agree with the referee that it would be helpful to systematically investigate the present materials using the same model. We now include data confirming that IDTT-C6-TIC, IDTT-C8-TIC, and IDTT-C10-TIC materials all show good self-spectral overlap; the Stokes shifts are smaller than 0.15 eV (Fig. S1 in the supplementary materials, Fig. R3 in the response letter), similar to IDIC.

Evaluating the energetic disorder through temperature dependent PL spectroscopy is somewhat difficult for these materials. Nevertheless, the preliminary data from the temperature dependent PL measurements indicate low energetic disorder of our molecular acceptors. Figure R3 in the response letter shows temperature-dependent PL spectra for IDTT-C8-TIC and IDTT-C10-TIC. This data reveals that the 0-0 PL peaks shift by only $<0.01\text{eV}$ and $<0.007\text{eV}$ for IDTT-C8-TIC and IDTT-C10-TIC, respectively, over a range of 250 degrees. This observation agrees with our previous study in FREAs, and can be explained by the high crystallinity of the IDTT-C8-TIC and IDTT-C10-TIC.

Figure R3. Temperature dependent PL of the drop casted IDTT-C6-TIC, IDTT-C8-TIC, and IDTT-C10-TIC acceptors

We prefer to follow up with these temperature-dependent PL measurements in a separate future publication about photophysical structure-function relations. The challenge that we are trying to overcome is that, so far, we could only obtain the temperature-dependent PL data for thick (drop-cast) films due to the detection sensitivity constraints. The PL spectra are likely affected by both inner filter effects, as well as morphology differences compared with thin films. Indeed, temperature-dependent PL measurements for IDTT-C6-TIC show a significant temperature-induced phase or aggregation behaviour change for thick films. Nevertheless, the small magnitude of peak shifts within each temperature-dependent series for IDTT-C8-TIC, and IDTT-C10-TIC supports the previous suggestion of low energetic disorder. Owing to the Covid-19 pandemic, our laboratories are supposed to reopen after seven months of closure. We look forward to carrying out a systematic study by expanding on these detailed photophysical aspects in a future project.

REVIEWERS' COMMENTS

Reviewer #2 (Remarks to the Author):

The authors provided detailed answers to all comments in my report- Most importantly, the characterization of the blend morphology is now presented and discussed in great detail. I recommend publication of this work as it is.

Reviewer #3 (Remarks to the Author):

I believe the manuscript improved after the revision considering reviewer comments suggestions and additional discussion. It is suitable to Nat Comm and the novelty is justifiable.

Reviewer #2 (Remarks to the Author):

The authors provided detailed answers to all comments in my report-Most importantly, the characterization of the blend morphology is now presented and discussed in great detail. I recommend publication of this work as it is.

Reply: We appreciate the reviewer for the very positive recommendation of our work.

Reviewer #3 (Remarks to the Author):

I believe the manuscript improved after the revision considering reviewer comments suggestions and additional discussion. It is suitable to Nat Comm and the novelty is justifiable.

Reply: We appreciate the reviewer for the very positive recommendation of our work.